# Study on measurement and prediction of agricultural product supply chain resilience based on improved EW-TOPSIS and GM (1,1)-Markov models under public emergencies

Hongzhi Wang[1], Li Lu [1]*, Zhaoli Liu[1], Yuxuan Sun[2]

**1** School of Economics and Management, Qingdao Agricultural University, Qingdao, Shandong, China,
**2** Michael Smurfit Graduate Business School, University College of Dublin, Dublin, Ireland

\* l13455386992@163.com

## Abstract

The COVID-19 pandemic, African Swine Fever, and other major public health emergencies have affected the agricultural product supply chain in recent years. It appeared in various chain breakdown and blocking issues, in which the resilience was drastically reduced and food security and social stability were greatly disrupted. This dissertation adopted an improved EW-TOPSIS method to evaluate resilience and determined the significance of influence factors of the agricultural product supply chain in China, showing that adjustment capability was closely connected to resilience. Through the empirical research on top listed enterprises (NHL, SQF, DBN, YILI, HTGF), it was found that the resilience of the industry was generally lower in 2020–2021 than in 2015–2019, and recovered and peaked in 2022. An improved Markov-modified GM (1,1) forecasting method was adopted to construct a resilience-predicting model. It was found that there would be a decline of resilience in 2024–2025, while a general growth with fluctuations trend was shown during the thirteen years before and after the breakout of the COVID-19 pandemic. In addition, this dissertation uses independent samples T-test and Solomon sensitivity analysis methods to verify the feasibility of the empirical results. Accordant enhancement mechanisms were proposed based on the empirical findings and results, which were expected to improve the risk-resistant capability of the domestic agricultural product supply chain under potential public emergency scenarios in the future. Our research findings can serve as a valuable reference for scientific decision-making and policy formulation to encourage the establishment of a robust agricultural product supply chain resilience system.

**Data availability statement:** We have publicly released all the data and relevant metadata that support the reported research findings in the ICPSR database within the Social Sciences section of PLOS ONE. The link for direct access to this data is: https://doi.org/10.3886/E222681V1.

**Funding:** The funding information is as follows: This research was supported by the Humanities and Social Science Research Project of Ministry of Education of China (Grant No. 22YJA790057). The project leader is Hongzhi Wang.

**Competing interests:** The authors have declared that no competing interests exist.

**Abbreviations:** CPC, Community Part of China; RCRC, Report of the Central Rural Work Conference; RWG Report on the Work of the Government; NHL New Hope Liuhe Co., Ltd; SQF Sanquan Food Co., Ltd. ; DBN DBN Group; YILI Inner Mongolia Yili Industrial Group Co., Ltd; HTGF Haitian Flavouring Foodstuff Co., Ltd.

## 1. Introduction

In recent years, under the influence of the COVID-19 pandemic, regional conflicts, geopolitics and other emergencies, chain interruption and blockage of the agricultural product supply chain have frequently occurred in China. The resilience of the agricultural product supply chain has been significantly reduced, which has brought potential risks to the stable supply of agricultural products and has aggravated the food security problem. "Enhancing the resilience and security of supply chains", "Fostering the cultivation of a holistic perspective on food" and "Building a diversified food supply system"were proposed in the 20th National Congress Report of the CPC [1]."Overcoming multiple adverse impacts such as more severe natural disasters, we must persistently strengthen the foundation of agriculture and promote the comprehensive revitalization of the countryside"was reiterated by the Central Rural Work Conference in 2023 [2]. "Grasping the production of food and important agricultural product, and strengthening the construction of emergency supply bases for food basket product" was emphasized by the 2024 China Central Document No. 1 [3]. "Promoting the optimization and upgrading of the industrial and supply chains, enhancing the resilience and competitiveness of the industrial and supply chains, effectively maintaining the security and stability of the industrial and supply chains, and supporting the smooth flow of the national economic cycle" was raised by the Report on the Work of Government in 2024 [4]. These have revealed that the effective provision of agricultural products not only serves as a pivotal assurance for food security but also constitutes a critical underpinning for realizing the holistic perspective on food.

Taking the COVID-19 pandemic as an example, various regions implemented preventive measures such as city lockdowns, road closures, and quarantines in China. On the one hand, the sales volume of agricultural products has significantly decreased. According to data from the Ministry of Agriculture and Rural Affairs of the People's Republic of China (MARA) (2021), the total value of China's agricultural product exports in 2020 was $76.03 billion, which has decreased by 3.23% than last year. On the other hand, a large volume of unsalable agricultural products became a crucial issue. According to the data from the People's Daily, there are 7.29 million tons of agricultural products are unsalable in the domestic market. During the peak time of the pandemic, more than 230 million people in China have been restricted at home, thus, the provision of daily necessities has become the most significant challenge in sustaining citizens' livelihoods.

Hence, during public emergencies such as COVID-19, the agricultural product supply chain faced unprecedented challenges. To ensure food safety and market stability, enhancing the resilience of the supply chain has become crucial, not only for China but for the entire world. This research work is grounded in the case facts of five leading Chinese agricultural enterprises (NHL, SQF, DBN, YILI, HTGF) from 2015 to 2022, optimizing the existing methods and proposing an improved EW-TOPSIS and GM (1,1)-Markov models to measure and predict resilience. The research goal is to explore the influencing factors of supply chain resilience in public emergencies, analyze the performance and development trends of these enterprises during the

pandemic, and provide insights and suggestions for the future development direction of agricultural supply chain enterprises. Based on these, the research work aims to address four key questions (RQs).

*RQ1.* What are the driving factors of agricultural product supply chain resilience under public emergencies?
*RQ2.* How are agribusinesses performing and their changing tends during the COVID-19 pandemic?
*RQ3.* How will agribusinesses develop in the coming future with the diminishing impact of the COVID-19 pandemic?
*RQ4.* How can agribusinesses enhance the resilience of agricultural product supply chains in similar public health emergency scenarios?

To address these problems, we established a resilience evaluation system based on fifteen key factors. Besides, the supply chain resilience of five sample enterprises was evaluated over eight years before and after the outbreak of COVID-19 (2015–2022), using the improved Entropy Weight (EW) method and the technique for order of preference by similarity to ideal solution (TOPSIS). Furthermore, a novel Markov-modified grey GM (1,1) forecast model was constructed to draw a clear tendency of resilience development for the next five years (2023–2027). All results and intermediate variables are listed explicitly in tables, or depicted in statistical figures, and their reliability and credibility are verified by adopting residual testing methods. The evaluation system, forecast model, and corresponding analysis further introduce the enhancement mechanisms of China's agricultural product supply chain resilience, which is proposed at the end of the dissertation. Fig 1 depicts the framework of this study.

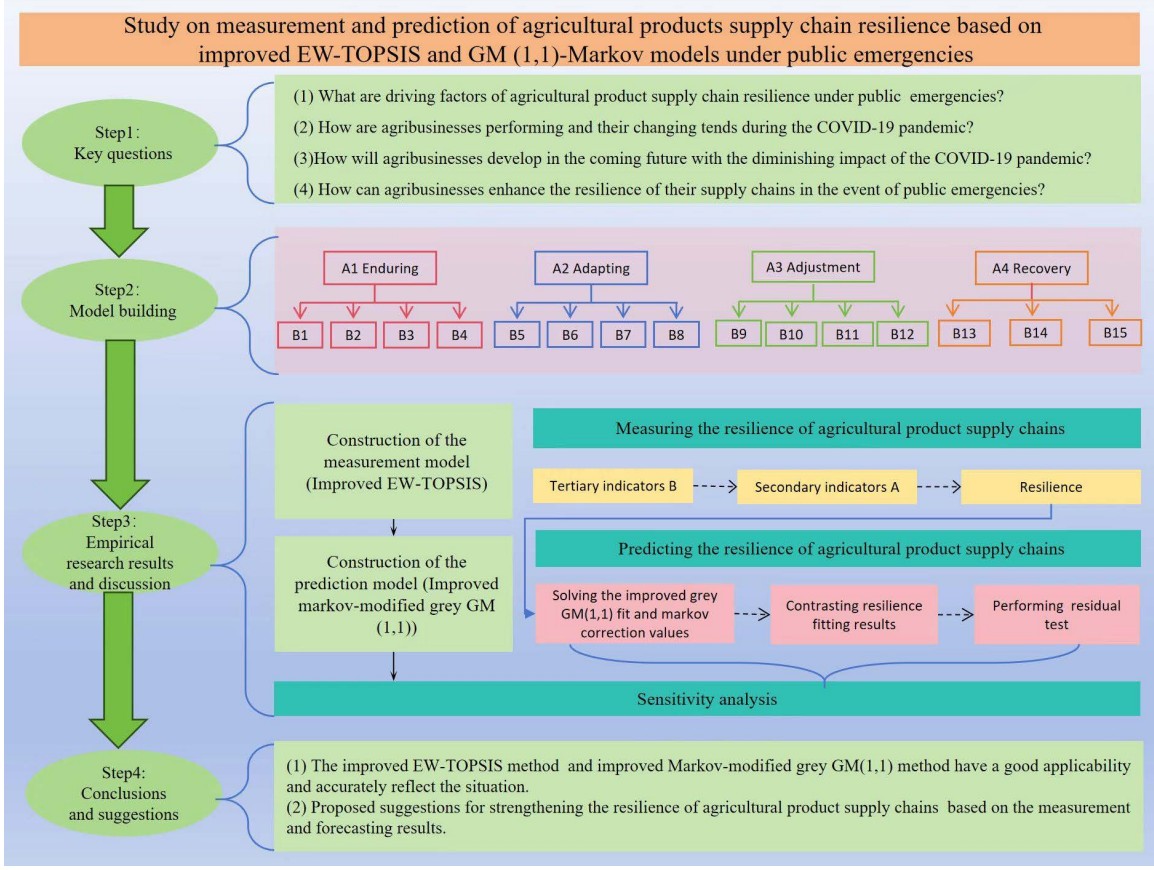

**Fig 1. Technical route map. Source: By the authors.**

Five sections comprise the study, where the first section discusses the impacts of agricultural product supply chain resilience and the necessity for them; section 2 summarizes the review of literature on determinants related to the agricultural product supply chain resilience and existing solution approaches; section 3 describes the data collection, case studies, methodologies; section 4 summarizes and discusses the empirical research results and suggestions; while section 5 addresses the conclusions, limitations, and prospects.

## 2. Literature review

### 2.1. Agricultural product supply chain resilience

The theory of resilience has been widely applied in supply chain management. Regarding the concepts related to supply chain resilience, Svensson et al. (2000) [5]argue that vulnerability is a primary characteristic of supply chain resilience. They define supply chain vulnerability as the random disturbances that affect the stable operation of supply chains, emphasizing the adverse consequences resulting from risk sources and risk drivers exceeding the support scope of risk mitigation strategies. This concept was introduced to China by Ning et al. (2004) [6]. Remko et al. (2020) [5] suggest that supply chains gradually develop adaptability after experiencing various risks. Therefore, resilient supply chains can help agricultural enterprises achieve high-quality operations and sustainable development during public emergencies.

In recent years, the global supply chain has faced significant uncertainties due to the impact of public emergencies, and disruptions at any link may pose a threat to the secure operation of the supply chain. To address this challenge, some leading scholars have focused on optimizing the supply chain using optimization approaches such as metaheuristic approaches. Wang et al. (2010) [7] adopted the objective optimal heuristic selection (OOHS) algorithm aiming to minimize the total cost of the supply chain, thereby significantly enhancing the market competitiveness of the supply chain. Peng et al. (2016) [8] proposed an enhanced bee colony algorithm for optimizing the agricultural product supply chain, and through robustness and computational efficiency analysis, validated the superiority of the algorithm in optimization effects. Guo et al. (2016) [9] combined genetic algorithms with optimization algorithms, designed a self-learning composite genetic algorithm, and achieved dual-stage comprehensive optimization for the construction of a green supply chain network, providing effective decision-making support for managers. Dong et al. (2022) [10] designed a two-stage heuristic algorithm and provided more effective solutions compared to CPLEX and particle swarm optimization algorithms. Yu et al. (2024) [11] developed an improved particle swarm optimization algorithm (HPSO-SA), which enhanced solution efficiency through specific heuristic rules to address the situation of core suppliers' disruptions. Additionally, in the field of agricultural product supply chain optimization research, relevant scholars also have focused on methods such as the collaborative quality control model for agricultural product supply chains [12], the collaborative quality control mechanism for agricultural product supply chains[13,14], and the delivery optimization model for agricultural product supply chains[15,16], and so on.

### 2.2. Food security and agricultural product supply chain resilience

According to relevant scholars' research, food security and the resilience of agricultural product supply chains are of great significance during public emergencies. The agricultural product supply chain is a critical link connecting agricultural producers and consumers, encompassing multiple stages such as production, processing, storage, transportation, wholesale, and retail. It serves as a bridge between farmers and the market, enabling the transition of agricultural products from the field to the table [17]. Numerous scholars agree that an efficient, reliable, and resilient supply chain is crucial for ensuring food security [18]. This not only guarantees the quality and safety of agricultural products but also makes basic food needs more affordable for more people by reducing costs and improving efficiency [19]. As the world's largest producer and importer of agricultural products, China has always attached great importance to the stable and secure supply of key agricultural products and food security. This is not only vital for national economic development and social stability but also crucial for national security and national self-reliance [20]. On one hand, the efficient operation of the agricultural product supply chain is not only an important prerequisite for ensuring national food security but also a significant driver of

national economic growth. On the other hand, as a major global market for agricultural products, China is deeply involved in international agricultural trade and supply chain cooperation, maintaining the balance of global food and agricultural product supply and demand, and contributing China's solutions to global food security and the achievement of sustainable development goals [21, 22]. Therefore, the foundation of food security lies in the security of the supply chain. Enhancing the risk resistance capability of the food supply chain is particularly important.

## 2.3. Measuring agricultural product supply chain resilience

The main perspectives of quantitative research work on agricultural product supply chain resilience include evaluation index systems, factor analysis, event models, optimization models, and so on. Among them, the evaluation system models have the most diversity. Zhou [23] constructed a fuzzy comprehensive performance evaluation model for the agricultural product supply chain by applying the evaluation method of AHP-FCE. Cheng and Li [24] used interval hierarchy analysis (IAHP) and fuzzy comprehensive evaluation (FCE) to score the performance of agricultural product supply chain enterprises. Qin and Zhang [25] utilized fuzzy cluster analysis (FSQCA) to study the influence mechanism and adaptability of agricultural product supply chain resilience. Zou et al. [26] constructed a risk assessment model for fresh produce based on the SCOR model. Yang and Jia [27] put forward the agricultural product supply chain coupling effect measurement method to study the coupling effect of the "smart sharing" agricultural product supply chain. Zhong et al. [28] proposed the supply chain resilience evaluation system mainly includes organizational vulnerability, structural robustness, rapid repair, learning innovation, and sustainable optimization. Fu [29] based on the operation mode of the fresh agricultural product supply chain in China, put forward the necessity of constructing a fresh agricultural product supply chain resilience evaluation index system. Lü et al. [30] utilized the stepwise weighted assessment and analysis ratio (SWARA) method, and analyzed supply chain risk controllability, adaptability, collaboration, and learning. Therefore, to explore the influencing factors of the agricultural product supply chain resilience, it is crucial to construct a resilience evaluation index system for agricultural product supply chains [31, 32].

In addition, in order to further improve the supply chain resilience and better satisfy customer demand. Alternate suppliers and remaining inventory were considered as resilience strategies [33]. Some studies found that fuzzy hierarchy analysis (AHP) and fuzzy prioritization technique (TOPSIS) methods to measure and analyze the drivers of supply chain resilience [34, 35].Relevant scholars based on the EW-TOPSIS method, using the improved TOPSIS method to rank the sample size, to achieve the purpose of effective quantitative evaluation of sample size [36–38]. Jain et al. [39] used the ISM model to study the influencing factors of supply chain resilience and explained the dynamic relationship between various drivers.

## 2.4. Predicting agricultural product supply chain resilience

The related research work on supply chain resilience prediction started late, and after the epidemic outbreak, scholars at home and abroad began to conduct a large number of related studies on it. Ma et al. [40] proposed to predict the annual grain production by establishing a GM(1,1) model and used the Markov model to improve the accuracy of the predicted value of grain production. Qi et al.[41] suggested that with the long-term epidemic of the COVID-19 pandemic, firms have become progressively more responsive and adaptive to risk, but until the end of 2022, resilience can't yet be restored to pre-epidemic levels. At the present stage, the prediction of the development trend of the agricultural product supply chain resilience was mostly based on the methods of grey system, Markov, time series, linear regression, and so on. Among them, the Grey-Markov model was more widely used. Some studies found that the model corrects the residuals of the Grey prediction through the Markov method, it can overcome the disadvantage of the single, monotonicity of the GM(1,1) prediction approach, and improve the sensitivity of the variables under the influence of uncertain factors, thus improving the accuracy of the prediction [40]. Relevant scholars optimized the residuals for grey prediction, proposed the Markov grey prediction model based on positive and negative residuals, and pointed out that the amount of data will have a large

impact on the predicted value, it provides a new way of thinking for optimizing the prediction model and improving the prediction accuracy [42].

In summary, domestic and abroad scholars have either established a variety of quantitative models from the perspective of resilience and coupling based on the application of theories about supply chains and agricultural traits, or applied plenty of forecast method to explore the pattern of change under recent environmental restraints. They also predicted the probability of risk occurrence in the coming years. A more detailed comparison is summarized in Table 1. However, fewer studies have focused on the overall assessment of evaluation of the agricultural supply chain resilience, especially lacking of clarifications of the development trend in the post-epidemic era. Moreover, effective enhancement mechanisms were deficient and lacked of the proposing of contemporary strategies on agricultural business sectors, which related to managing emergency countermeasures and decision-making processes. Therefore, this research will take this as a starting point to fill in the gaps that exist in the current literature.

## 3. Methodology

### 3.1. Introduction of methodology

This study analyzes the factors affecting the resilience of agricultural product supply chains and their development trends over the next five years by collecting both qualitative and quantitative data. Two mathematical models were used to further determine strategies for enhancing the resilience of agricultural product supply chains under potential public emergencies. On one hand, an indicator evaluation system was established, and the improved EW-TOPSIS method was used to assess the supply chain resilience of agricultural enterprises. On the other hand, a prediction model was established, employing the improved Markov-corrected GM (1,1) method to forecast the development trends over the next five years (Qi et al., 2022; Hao, 2023).

**Table 1. Research contribution and methods used in existing literature.**

| Authors | Reference | Research Contribution | Used Methods |
|---|---|---|---|
| Wang et al., (2010)<br>Peng et al. (2016)<br>Guo et al. (2016) | [7]<br>[8]<br>[9] | Minimize the total cost of the supply chain<br>Obtain the Pareto optimal solution for SC<br>Achieve dual-stage comprehensive optimization for SC | OOHS algorithm<br>Bee colony algorithm<br>Composite genetic algorithm |
| Dong et al. (2022) | [10] | Minimize logistics costs of SC | Heuristic algorithm |
| Yu et al. (2024) | [11] | Address the situation of core suppliers' disruptions | Particle swarm optimization algorithm (HPSO-SA) |
| Zhou (2020) | [23] | Constructs a fuzzy comprehensive performance evaluation model of SC | AHP-FCE method |
| Cheng et al. (2021] | [24] | Evaluate the performance of agricultural product supply chain enterprises | ROF model |
| Qin et al. (2021) | [25] | Analyze the influence mechanism and adaptability of SCR | Fuzzy cluster analysis (FSQCA) method |
| Zou et al. (2021) | [26] | Construct a risk assessment model for fresh produce | SCOR model |
| Yang et al. (2021)<br>Zhang et al. (2024) | [27]<br>[32] | Study the coupling effect of "smart sharing" SCR<br>Evaluate risk management of green supply chains for agricultural products | Coupling effect measurement method<br>Social network evaluation framework |
| Sun et al. (2023) | [43] | Evaluate for the sustainable development of fresh agricultural products logistics enterprises | AHP and empowerment-TOPSIS method |
| Jain et al. (2017) | [39] | Explain the dynamic relationship of SC | ISM model |
| Qi et al. (2022) | [41] | Propose early warns for SCR | Grey prediction model |
| Ma et al. (2020) | [40] | Predict agricultural product yields accurately | Grey-Markov model |

## 3.2 .Case study

After the theoretical analysis in the previous sections, the improved EW-TOPSIS model and the GM (1,1)-Markov model demonstrated good performance in the measurement and prediction of resilience in agricultural supply chains. To verify their practicality and effectiveness, a case study was conducted on five leading Chinese agricultural listed companies. The production bases and supply chain networks of these companies cover the entire country (such as NHL in Chengdu, Sichuan; SQF in Zhengzhou, Henan; DBN in Beijing; YIII in Hohhot, Inner Mongolia; and HTGF in Foshan, Guangdong), located in regions rich in agricultural resources, which can reflect the characteristics of supply chain resilience in different areas of China. Moreover, the COVID-19 pandemic from 2019 to 2022 had a profound impact on the global economy, causing frequent disruptions and bottlenecks in agricultural supply chains. Researching this issue is of great practical significance for ensuring the supply of agricultural products and food security. The main growth parameters of the supply chains of these five companies are shown in Table 2.

After determining the weights using the Improved Entropy-TOPSIS method, this dissertation combines Solomon's examples to evaluate the impact of Inventory Turnover Rate (ITR), Supply Chain Operating Cost (SCOC), and Supplier Spatial Distance (SSD) on supply chain resilience through sensitivity analysis [44]. The following steps can be taken to achieve this:

**Step 1:** constructing a supply chain resilience evaluation function.

$$R = W_{ITR} * ITR + W_{SCOC} * SCOC + W_{SSD} * SSD \tag{1}$$

Where represents the weight of inventory turnover ratio, represents the weight of supply chain operating cost, and SSD represents the weight of supplier spatial distance. ITR, SCOC, and SSD indicate the standardized values.

**Step 2:** Calculating the partial derivatives of supply chain resilience (R)with respect to each parameter to assess the impact of each parameter on supply chain resilience.

$$\Delta R_{ITR} = R(ITR + \Delta ITR, SCOC, SSD) - R(ITR, SCOC, SSD) \tag{2}$$

$$\Delta R_{SCOC} = R(ITR, SCOC + \Delta SCOC, SSD) - R(ITR, SCOC, SSD) \tag{3}$$

$$\Delta R_{SSD} = R(ITR, SCOC, SSD + \Delta SSD) - R(ITR, SCOC, SSD \text{ (4)} \tag{4}$$

**Step 3:** Calculating Sensitivity.

Inventory turnover rate sensitivity calculation formula:

**Table 2. Information of five agricultural product listed companies from 2015 to 2022.**

| Sample | Location | Transportation method | Spatial distance | Inventory turnover rate | Transportation cost | Storage cost |
|--------|----------|----------------------|------------------|------------------------|---------------------|--------------|
| NHL | Eastern | road | 500-800 | 6.500 | 500 | 60 |
| SQF | Central | road or railway | 300-500 | 7.500 | 420 | 50 |
| DBN | Central | road | 200-400 | 5.500 | 400 | 55 |
| YILI | Western | road | 300-600 | 8.000 | 450 | 55 |
| HTGF | Eastern | road | 100-300 | 7.000 | 350 | 50 |

Note: The unit for spatial distance is (km), the unit for inventory turnover rate is (times/year), the unit for transportation cost is (yuan/kg-km), and the unit for storage cost is (yuan/month).

$$S_{ITR} = \frac{\Delta R_{ITR}}{\Delta ITR}$$

(5)

Supply chain operating cost sensitivity calculation formula:

$$S_{SCOC} = \frac{\Delta R_{SCOC}}{\Delta SCOC}$$

(6)

Supplier spatial distance sensitivity calculation formula:

$$S_{SSD} = \frac{\Delta R_{SSD}}{\Delta SSD}$$

(7)

When is large, it indicates that the inventory turnover rate has a significant impact on supply chain resilience. If is large, it indicates that supply chain operating costs have a significant impact on supply chain resilience. If is large, it indicates that the supplier's delayed delivery rate has a significant impact on supply chain resilience.

Through sensitivity analysis methods, the impact of supply chain growth parameters on supply chain resilience can be quantified, thereby providing decision-making support for the optimization of agricultural product supply chains.

### 3.3. Data sourcing and processing method

This article is based on the research background of the COVID-19 pandemic, selecting five typical agricultural product listed companies in China (NHL, SQF, DBN, YIII, and HTGF) as research samples. The empirical study is conducted over an 8-year period before and after the outbreak of the COVID-19 pandemic (2015–2022). The selected agricultural product listed companies are categorized according to the "Industry Classification Guidelines for Listed Companies (Revised in 2012)" issued by the China Securities Regulatory Commission. On the one hand, the data of objective quantitative indicators mainly come from the CSMAR database and the annual reports of the agricultural product listed companies. On the other hand, imitating the method from Mena et al. (2022) and Rai et al. (2021) Subjective qualitative indicators data are collected from a questionnaire survey that involved managers from listed companies, professionals from the industry, and scholars from the relevant field. They scored samples' performance under various subjective indicators at an integer from 1 to 5. The final value of each record is the mean of questionnaire scores. Sample data from 2015 to 2022 are all collected in terms of years. To ensure the reliability of the empirical results and to eliminate the impact of extreme values, all continuous variables are winsorized at the 1% level from both the top and bottom. The software used for the research includes Excel 2016 and PYTHON.

### 3.4. Modelling method

#### 3.4.1. Improved the EW-TOPSIS method.

The improved Entropy-TOPSIS model is an objective weight allocation model that reflects the amount of information contained in evaluation indicators based on the degree of influence of their values, thereby calculating the information entropy of each indicator. Compared with other evaluation models, the improved entropy method can objectively reflect the importance of each indicator, effectively reduce the subjectivity of data, and better meet the needs of empirical operations. In addition, the TOPSIS (Technique for Order Preference by Similarity to Ideal Solution) method primarily determines the superiority of plans based on the ideal solution and negative ideal solution of the evaluation object. It has the advantages of simple calculation, broad applicability, and large sample size, providing moderate and reasonable evaluation results.

This research work uses the improved Entropy-TOPSIS model to measure the weight of indicators for the resilience of agricultural product supply chains. First, data standardization is performed, and the improved entropy method is used

to assign weight values to each indicator in the agricultural product supply chain. Then, through quantitative ranking, the TOPSIS method is applied to evaluate the resilience of the supply chains of listed agricultural product companies. The improved Entropy-TOPSIS model not only considers the correlation between various influencing factors within listed agricultural product companies but also accurately measures the impact degree of different subjective and objective indicators on the resilience of the agricultural product supply chain and determines their indicator weights, thereby improving the accuracy and objectivity of the evaluation results.

Since the traditional entropy method is prone to cause too small or too large indicator weight when the data distribution is too dense or dispersed, this research work introduces the Z-score method to improve the processing of the initial sample data [45], so that the sample data are normally distributed in [0,1] to overcome the phenomenon of too dense or too sparse data. Among them, the Z-score model is as follows:

$$z_{ij} = \frac{x_{ij} - \overline{x_j}}{s_j}$$

(8)

Where is the mean of the j indicator data; is the number of indicators; is the standard deviation of the j indicator data.

The TOPSIS method evaluates the sample's quality of performance by introducing absolute positive and negative ideal solutions (Tang, 2019). A well-performed sample numerically approaches the absolute positive solution under major indicators. Since the traditional TOPSIS method ignores the primary and secondary differences of each indicator and the influence of indicator weight, this research work utilizes the Z-score to improve the indicator weight accuracy, which makes the TOPSIS calculation take into account the influence of the weight on the score. As a result, the weighted Euclidean distance calculation method of the two ideal solutions of the sample distance in the TOPSIS is calculated as follows: Where indicates an absolute positive ideal solution for data under indicator $> j$; indicates an absolute negative ideal solution for data under indicator j

**3.4.2. Improved GM (1,1)-Markov combination prediction method.** The grey forecast method is used to explore the development tendency of a series by building a grey differential equation (Yu et al., 2020). GM (1,1) method builds a first-order differential equation from an explanatory sequence (Rajesh, 2016) to implement the prediction function. Compared to other predictive models, this model not only reflects the general overall trend of the sample based on quantified resilience levels, but also has the advantages of improving prediction accuracy, being suitable for short-term predictions, and being able to correct predicted values.

The traditional GM (1,1) is a model for constructing a first-order differential equation for grey prediction. The prediction method is based on establishing a grey differential equation for a known sequence and utilizing the method of least squares to find out the pattern of the sequence change. This treatment may lead to the volatility and randomness of the original data being ignored, thus affecting the prediction accuracy of the model. Drawing on the idea of improving GM(1,1) by scholars Ding and Zhou [46], this research work adopted the geometric mean method to smooth and improve the treatment of the sequence, and then the Markov model was used to correct the predicted value of the GM(1,1)to obtain the accurate prediction results.

The specific steps to improve the GM (1,1)-Markov combination prediction method are as follows:

$$d_i^+ = \sqrt{\sum_{j=1}^{n} w_j \left( z_j^+ - z_{ij} \right)^2}$$

(9)

$$d_i^- = \sqrt{\sum_{j=1}^{n} w_j \left( z_j^- - z_{ij} \right)^2}$$

(10)

**Step 1:** Constructing the initial grey prediction sequence.

$$s^{(1)}(t) = \begin{cases} s^{(0)}(t), & , t = 1 \\ \sqrt{\left[s^{(0)}(t)\right]^2 + \left[s^{(1)}(t-1)\right]^2}, & t > 1 \end{cases}$$

(11)

**Step 2:** Running the GM (1,1) model to fit the constructed sequence to obtain a sequence of predicted values.
**Step 3:** Performing the differential reduction on the sequence of predicted values to obtain the true predicted value.

$$\hat{s}^{(0)}(t) = \sqrt{\left[\hat{s}^{(1)}(t)\right]^2 - \left[\hat{s}^{(1)}(t-1)\right]^2}$$

(12)

Where indicates the sequence of predicted value after smoothing at moment $t$; $\hat{s}^{(0)}(t)$ indicates the predicted value at moment $t$.
**Step 4:** Calculating the residual value of the GM (1,1) prediction.

$$\varepsilon(t) = \hat{s}^{(0)}(t) - s^{(0)}(t), t = 1, 2, \ldots, m$$

(13)

Where indicates the absolute residual error, indicates the predicted sequence.
**Step 5:** The GM (1,1) predicted values are tested for the magnitude of the parameters, such as the grey correlation R, the posterior variance ratio C, and the small probability error P. The grey correlation of the predicted sequence element is calculated as follows.

$$\eta_t = \frac{\varepsilon_{\min}(t) + \rho\varepsilon_{\max}(t)}{\varepsilon(t) + \rho\varepsilon_{\max}(t)}$$

(14)

When the average value of the grey correlation $\eta_t$ of each grey correlation prediction series R>0.6, the grey correlation test of the model is generally considered to have passed.
Drawing on the supply chain resilience scoring criteria of related scholars [47], the test criteria of the posterior variance ratios and small probability errors are shown in Table 3.
**Step 6:** The residual values computed in Step 4 are divided into four state classes with consecutive intervals, and the sequence of classes is a Markov prediction sequence. The state rank t corresponding to each value of the residual sequence $\varepsilon(t)$ is divided as follows:

$$e_{1k} \leq \varepsilon(t) < e_{2k}, k = 1, 2, 3, 4.$$

(15)

Where $e_{1k}$ indicates lower bound of rank$E_k$ ; $e_{2k}$ indicates upper bound of rank$E_k$.
**Step 7:** Weighting and summing rank$E_k$ to obtain P'(k) is calculated as follows:

**Table 3. Posterior parameter level comparison.**

| P | C | Effect |
|---|---|---|
| $0.95 \leq P \leq 1.00$ | $0.00 \leq C < 0.35$ | better |
| $0.80 \leq P < 0.95$ | $0.35 \leq C < 0.50$ | good |
| $0.70 \leq P < 0.80$ | $0.50 \leq C < 0.65$ | general |
| $0.00 \leq P < 0.70$ | $0.65 \leq C \leq 1.00$ | bad |

$$P'(k) = \sum_{k=1}^{m} w_k P^{(k)}$$

(16)

Where $w_k$ is the autocorrelation coefficient, it is calculated as follows:

$$w_k = \frac{p_i^{(k)}}{\sum_{i=1}^{n} p_i^{(k)}}$$

(17)

**Step 8:** Calculating the state probability matrix:

$$\pi_j(k) = \sum_{i=1}^{3} \pi_i(k-1) P'_j, j = 1, 2, 3, 4.$$

(18)

Where $\pi_i(k)$ indicates the probability that state $E$ is generated at the $K$ step of the transfer.

**Step 9:** The recursive formula that represents the state probability matrix after the $K$ step transfer is as follows:

$$\pi(k) = \pi(k-1) \boldsymbol{P'}(k)$$

(19)

**Step 10:** The Markov model is applied to correct the sequence of GM (1,1) predicted value and the residual sequence value, and the Markov correction formula is as follows.

$$\hat{s}_r^{(0)}(t) = \hat{s}^{(0)}(t) \pm \frac{e_{1j} + e_{2j}}{2}$$

(20)

Where $\hat{s}_r^{(0)}(t)$ indicates Markov's corrected prediction at the moment $t$.
Where the correction sign ($\pm$) is judged:

$$\begin{cases} +, \ \hat{s}^{(0)}(t) < s^{(0)}(t) \\ -, \ \hat{s}^{(0)}(t) > s^{(0)}(t) \end{cases}, t = 2, \ 3, \ldots, m.$$

(21)

When $t > m$, it is necessary to estimate the sign of the correction of the sequence to be predicted since it has no corresponding value.

**Step 11:** For calculating the predicted value for future moments, it is necessary to construct a Markov prediction sequence from the known correction symbols. Repeating steps 13-step 17 for the estimation of the correction symbols and the correction of the predicted value for the future moments.

## 4. Results and discussion

### 4.1. Agricultural product supply chain resilience evaluation result.

**4.1.1. Detailed indictors list of resilience evaluation system.** Agricultural products have the characteristics of being easy to damage, easy to deteriorate, difficult to store, and difficult to transport, and their supply chain construction is large-scale, with high professional requirements, higher fixed costs, and time-sensitive requirements. At the same time, the yield of agricultural products changes more strongly with the season, climate, and other natural factors, and the storage link of logistics is more demanding. Therefore, the selection of resilience evaluation indexes of agricultural product supply chains should consider both the special characteristics of agricultural products and the impact of the external environment on the supply chain.

To scientifically, reasonably and accurately evaluate the resilience of the agricultural product supply chains, this research work is based on the status of China's agricultural product supply chain during the COVID-19 pandemic, and through reviewing the literature [48–50] and field investigation. Therefore, this research work establishes a four-level evaluation index to measure the resilience of agricultural product supply chains under public emergencies, as shown in Table 4.

In Table 4, enduring ability refers to the ability of the agricultural product supply chain to resist pressure in the face of public emergencies, which is also the ability to guarantee the security of the agricultural product supply chain. Adaptability refers to the ability of an agricultural product supply chain to respond and adjust effectively to the challenges of the internal and external environment to ensure the agricultural product supply chain remains stable. On the one hand, it enables the agricultural product suppliers to flexibly adjust the purchasing amount to meet the market demand. On the other hand, it optimizes the logistic operation and swiftly adjusts the level of inventory. Adaptability is the ability of agricultural product suppliers to identify new opportunities, mitigate the impact of unexpected events, and reorganize their supply chains in a timely and accurate manner. Recovery ability refers to the ability of agricultural product suppliers to quickly and effectively return to normal operation after being hit by emergencies, emphasizing that each node of the agricultural product supply chain can coordinate the production links as soon as possible, to ensure the normal operation of agricultural products' production, warehousing, transportation and sales.

**4.1.2. The results of the weight calculation for the tertiary indicators.** The data of related indicators officially released by five listed agricultural product companies from 2015 to 2022 are normalized, and Equation (8) in the improved entropy method is applied to homogenize the initial series, and the formula is used to calculate the weight of each indicator, to obtain the tertiary indicator weight matrix W={0.050, 0.053, 0.050, 0.050, 0.039, 0.080, 0.106, 0.060, 0.031, 0.079, 0.062, 0.140, 0.065, 0.061, 0.076}. After eliminating invalid information, the information entropy value of the tertiary indexes, the information utility value, and the weight coefficients are shown in Table 5.

**4.1.3. The results of the weight calculation for the secondary indicator.** The weight of the above tertiary indicators is summed to obtain the secondary indicator weight, as shown in Table 6.

Drawing results from Table 5 and Table 6, at the level of secondary, the "enduring ability" accounted for 20.30%, the "adaptability" accounted for 28.50%, the "adjustment ability" weighted 31.10% and the "recovery ability" accounted for

**Table 4. Index system for measuring the resilience of agricultural product supply chain.**

| Level 1 indicator | Level 2 indicator | Level 3 indicator | Direction |
|---|---|---|---|
| Agricultural product supply chain resilience (R) | Enduring ability of agricultural product supply chain($A_1$) | Marketing capabilities($B_1$) | + |
| | | Net profit($B_2$) | + |
| | | Gearing ratio($B_3$) | − |
| | | Administrative expenses($B_4$) | − |
| | Adaptability of agricultural product supply chain($A_2$) | Suppliers' concentration($B_5$) | + |
| | | Inventory turnover ratio($B_6$) | + |
| | | Suppliers' cash flow ratio($B_7$) | + |
| | | Suppliers' revenue($B_8$) | + |
| | Adjustment ability of agricultural product supply chain($A_3$) | Supply chain operation cost($B_9$) | − |
| | | Level of digital technology($B_{10}$) | + |
| | | Agribusiness collaboration efficiency($B_{11}$) | + |
| | | Delayed delivery rate($B_{12}$) | − |
| | Recovery ability of agricultural product supply chain($A_4$) | Risk management level($B_{13}$) | + |
| | | Supply chain network complexity ($B_{14}$) | + |
| | | R&D expenses ($B_{15}$) | + |

**Table 5. Level 3 indicator calculation results.**

| Level 3 indicator | Information value | Information utility value | Weighting coefficient |
|---|---|---|---|
| $B_1$ | 0.965 | 0.035 | 0.050 |
| $B_2$ | 0.963 | 0.037 | 0.053 |
| $B_3$ | 0.965 | 0.036 | 0.050 |
| $B_4$ | 0.965 | 0.035 | 0.050 |
| $B_5$ | 0.972 | 0.028 | 0.039 |
| $B_6$ | 0.943 | 0.057 | 0.080 |
| $B_7$ | 0.925 | 0.075 | 0.106 |
| $B_8$ | 0.958 | 0.042 | 0.060 |
| $B_9$ | 0.979 | 0.021 | 0.031 |
| $B_{10}$ | 0.944 | 0.056 | 0.079 |
| $B_{11}$ | 0.956 | 0.044 | 0.062 |
| $B_{12}$ | 0.901 | 0.099 | 0.140 |
| $B_{13}$ | 0.954 | 0.046 | 0.065 |
| $B_{14}$ | 0.957 | 0.043 | 0.061 |
| $B_{15}$ | 0.947 | 0.053 | 0.076 |

**Table 6. Ranking of level 2 indicator weight.**

| Level 2 indicator | $A_1$ | $A_2$ | $A_3$ | $A_4$ |
|---|---|---|---|---|
| Weight | 0.203 | 0.285 | 0.311 | 0.201 |
| Sort | 3 | 2 | 1 | 4 |

20.10%, respectively. At the level of the tertiary, delayed delivery rate, suppliers' cash flow ratio, and inventory turn-over ratio accounted for the highest proportion of weight framework with over 32.6% total. These significant indicators reflected that the interplay of node enterprises in the supply chain plays an important role in the resilience system. External conditions such as supplier operational efficiency, upstream supply, and downstream demand are indispensable for maintaining the resilience level. Although the lockdown and quarantine have stroked the agricultural product supply chain since the outbreak of the COVID-19 pandemic, the pandemic impact indicator accounted for slight significance in the system. The first research question Q1 was thereby systematically answered after completing the weighting indicator system.

**4.1.4. The evaluation score reflects the resilience level of sample enterprises.** Equations (9) and (10) in the TOPSIS comprehensive evaluation method are utilized for the calculation of the agricultural product supply chain resilience score. Referring to the theories of related scholars on the evaluation criteria of the agricultural product supply chain resilience, as shown in Table 7 [51]. The agricultural product supply chain resilience of five listed agricultural products companies from 2015 to 2022 is scored and divided into grades as shown in Table 8.

Results in Table 8 reflected that the resilience score of major samples is at the "Moderate" level. There was only a "Higher" level which is the score of the NHL in the year 2022. However, only the "Lower" level is evaluated by the NHL in 2019 and 2020. No "Higher" level was classified in any sample company or year. A significant reason that could be conducted is that the profitability and financial status of most sample companies in 2022 is much better than in previous years, drawing from the data in financial reports (Eastmoney Securities, n.d.). Another reason was reflected in the questionnaire survey. The given scores of subjective indicators (Risk management, Supply chain network complexity, et cetera) in the year 2019–2021 were concentrated between 1–3. Lower values under the above significant indicators consequently resulted in lower evaluation scores of resilience level for such samples.

**Table 7. Evaluation criteria for the resilience of agricultural product supply chain.**

| Rating levels | Sort | Limitation resilience score |
|---|---|---|
| I | Lower | $0.00 \leq s < 0.20$ |
| II | Low | $0.20 \leq s < 0.40$ |
| III | Medium | $0.40 \leq s < 0.60$ |
| IV | Higher | $0.60 \leq s < 0.80$ |
| V | Highest | $0.80 \leq s < 1.00$ |

**Table 8. 2015-2022 supply chain resilience scores for five agricultural product companies.**

| Year | Rank | Sample | Score | Level | Year | Rank | Sample | Score | Level |
|---|---|---|---|---|---|---|---|---|---|
| 2015 | 1 | DBN | 0.420 | Moderate | 2019 | 1 | HTGF | 0.571 | Moderate |
| 2015 | 2 | HTGF | 0.415 | Moderate | 2019 | 2 | DBN | 0.474 | Moderate |
| 2015 | 3 | NHL | 0.328 | Lower | 2019 | 3 | SQF | 0.442 | Moderate |
| 2015 | 4 | SQF | 0.269 | Lower | 2019 | 4 | YILI | 0.436 | Moderate |
| 2015 | 5 | YILI | 0.244 | Lower | 2019 | 5 | NHL | 0.387 | Lower |
| 2016 | 1 | HTGF | 0.451 | Moderate | 2020 | 1 | DBN | 0.536 | Moderate |
| 2016 | 2 | SQF | 0.377 | Lower | 2020 | 2 | HTGF | 0.509 | Moderate |
| 2016 | 3 | NHL | 0.348 | Lower | 2020 | 3 | SQF | 0.465 | Moderate |
| 2016 | 4 | YILI | 0.347 | Lower | 2020 | 4 | YILI | 0.415 | Moderate |
| 2016 | 5 | DBN | 0.286 | Lower | 2020 | 5 | NHL | 0.376 | Lower |
| 2017 | 1 | HTGF | 0.530 | Moderate | 2021 | 1 | SQF | 0.574 | Moderate |
| 2017 | 2 | DBN | 0.399 | Lower | 2021 | 2 | HTGF | 0.572 | Moderate |
| 2017 | 3 | NHL | 0.358 | Lower | 2021 | 3 | YILI | 0.560 | Moderate |
| 2017 | 4 | SQF | 0.341 | Lower | 2021 | 4 | DBN | 0.532 | Moderate |
| 2017 | 5 | DBN | 0.295 | Lower | 2021 | 5 | NHL | 0.402 | Moderate |
| 2018 | 1 | HTGF | 0.590 | Moderate | 2022 | 1 | NHL | 0.721 | Higher |
| 2018 | 2 | NHL | 0.442 | Moderate | 2022 | 2 | SQF | 0.543 | Moderate |
| 2018 | 3 | SQF | 0.400 | Moderate | 2022 | 3 | HTGF | 0.504 | Moderate |
| 2018 | 4 | YILI | 0.382 | Lower | 2022 | 4 | DBN | 0.496 | Moderate |
| 2018 | 5 | DBN | 0.342 | Lower | 2022 | 5 | YILI | 0.488 | Moderate |

Draw a trend chart according to the result of Table 8, showing as follows:

As can be seen from Table 7, Table 8, and Fig 2, the supply chain resilience scores of the five agricultural product companies from 2015 to 2022 are generally concentrated in the low to moderate range. The overall trend of the five agricultural product companies' resilience is down and then up: the supply chain resilience score drops sharply from 2019 to 2020 until it begins a slow rise in 2021, peaking in 2022. The overall trend indicates that after the supply cut-off crisis, the supply chain is restored and better than in 2018 and 2019 before the epidemic, and resilience is improved during the public emergency crisis event.

The potential reasons are as follows. From 2019 to 2021, the domestic epidemic is severe, and the sealing and control measures are strict in various places. In 2020, due to the impact of the epidemic's high-intensity, high-frequency, and wide-area outbreaks, there were varying degrees of production reductions and shutdowns in various production areas, and the logistics and transportation were blocked and stagnant. Enterprises have not recovered from the crisis, and the supply chain resilience of each sample is the lowest; by 2022, benefiting from the government's supply policy and the strengthening of the enterprise's internal adaptation and adjustment capacity, the threat of public emergencies to the

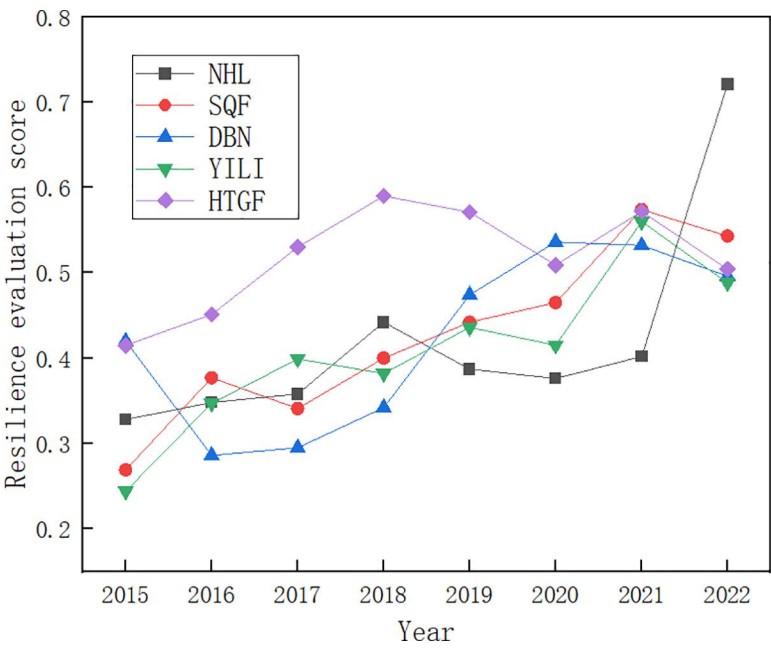

**Fig 2. Agricultural product supply chain resilience evaluation score.**

agricultural product supply chain resilience continues to be eliminated. So that the resilience of the agricultural product supply chain can be improved, and it can effectively cope with the impacts caused by public emergencies.

### 4.2. Agricultural product supply chain resilience forecast result

**4.2.1. The validation results of the resilience prediction fitting model.** To explore the development trend of agricultural product supply chain resilience and reduce the impact of emergencies on the agricultural product supply chain, this dissertation adopts the improved GM (1,1)-Markov combination prediction method for agricultural product supply chain resilience prediction. The accuracy of the improved prediction method needs to be verified. Based on the resilience evaluation, this research work continues to validate the fit of the agricultural product supply chain resilience using five typical data from 2015 to 2022.

**4.2.2. Calculating for improved GM (1,1) fit and residual value.** Based on the results of the supply chain resilience measurement of five listed agricultural product companies, the measured value in 2015 is taken as the initial sequence, the grey prediction sequence is constructed, and equations (11) and (12) are used to calculate the improved GM (1,1) fitting value, and the measured value of resilience during 2016–2022 are taken as the actual resilience reference value, and 2016–2022 five listed agricultural product companies' supply chain resilience is predicted and fitted, and the results are shown in Table 9.

Equation (13) is used to calculate the grey fitted residual value $\varepsilon(t)$ for the period 2016–2022, and the results are shown in Table 10.

As can be seen from Table 9, the residual values of each sample are mainly distributed within (0.001, 0.210), and the values are uneven, indicating that the fitting effect is not uniform and precise enough. Therefore, this dissertation continues to use the Markov model to correct the prediction results of the improved GM (1,1) to achieve more precision.

**4.2.3. Solving the Markov correction prediction fitted value.** The Markov hierarchical state Equation (15) is used for residual state hierarchy, the frequency of each state hierarchy is counted, the state transfer matrix is constructed and

**Table 9. 2016-2022 fitting value of GM (1,1) for supply chain resilience.**

| Sample | 2016 | 2017 | 2018 | 2019 | 2020 | 2021 | 2022 |
|---|---|---|---|---|---|---|---|
| NHL | 0.332 | 0.363 | 0.394 | 0.425 | 0.456 | 0.486 | 0.515 |
| SQF | 0.342 | 0.378 | 0.413 | 0.449 | 0.483 | 0.517 | 0.550 |
| DBN | 0.274 | 0.323 | 0.374 | 0.424 | 0.474 | 0.522 | 0.567 |
| YILI | 0.350 | 0.377 | 0.405 | 0.432 | 0.459 | 0.485 | 0.511 |
| HTGF | 0.510 | 0.517 | 0.524 | 0.531 | 0.538 | 0.545 | 0.552 |

**Table 10. 2016-2022 fitting residuals of GM (1,1) for supply chain resilience.**

| Sample | 2016 | 2017 | 2018 | 2019 | 2020 | 2021 | 2022 |
|---|---|---|---|---|---|---|---|
| NHL | 0.016 | 0.005 | 0.048 | 0.038 | 0.080 | 0.084 | 0.206 |
| SQF | 0.035 | 0.037 | 0.013 | 0.007 | 0.018 | 0.057 | 0.007 |
| DBN | 0.012 | 0.028 | 0.032 | 0.050 | 0.062 | 0.010 | 0.071 |
| YILI | 0.003 | 0.022 | 0.023 | 0.004 | 0.044 | 0.075 | 0.023 |
| HTGF | 0.059 | 0.013 | 0.066 | 0.040 | 0.029 | 0.027 | 0.048 |

Note: Residual value is in units of.

recursions are performed to find $\pi(2)$, $\pi(3)$,..., $\pi(k)$, the hierarchy of residual states after k-1 moments, as shown in Table 11.

Using Equations (16), (17), (18), (19), and (20), the fitted value of resilience prediction with Markov correction as well as the fitted residuals are calculated and the sign of the correction is determined according to Equation (21). The results of which are shown in Table 12 and Table 13.

**4.2.4. Contrasting resilience fitting results.** The fitted and average residual value of agricultural product supply chain resilience is calculated by the improved GM (1,1) model and the Markov-corrected prediction model. This provides a good answer to Q3 about the changing trends in agricultural product supply chain firms, as shown in Fig 3.

As can be seen from Fig 3, the fitted value of resilience obtained by the improved GM (1,1) deviates significantly from the reference series, while the fitted value of resilience after Markov correction fits better with the reference series. In addition, comparison of the average residual value of the fitted agricultural product supply chain resilience for five produce companies from 2015 to 2022. GM (1,1) modified values are calculated as R1= {0.060, 0.022, 0.033, 0.024, 0.035}, and the Markov corrected residuals are calculated as R2= {0.017, 0.004, 0.006, 0.008, 0.005}. It can be concluded that the total average residuals of the improved GM (1,1) fit value is 0.035, while the total average residuals of the Markov correction is 0.008, which is much smaller than the total average residuals of the GM (1,1) fit value. Therefore, this research work indicates that the improved GM (1,1)-Markov combination prediction method is more accurate.

### 4.3. Model verification and significance testing

**4.3.1. Qualitative analysis of model reliability.** In summary of the above results, the predicted sequences of the GM (1,1) model showed similarities as monotonous growth, while the fitting effect was not excellent enough. On the one hand, due to the small amount of data set, the prediction lacked accuracy; On the other hand, the GM (1,1) method had a poor fitness of oscillation sequences. After being modified by Markov correction, the altogether model overcame the defect of preliminary predicting by the grey system, and the predicted values were much closer to the original values. Moreover, the residuals were significantly minimized by the modification, and the fitness of the model achieved a higher level. Section 4.3.2 would demonstrate a more specific analysis of residuals.

 

**Table 11. 2016-2022 agricultural product supply chain resilience Markov sequence fitting state level.**

| Sample | 2016 | 2017 | 2018 | 2019 | 2020 | 2021 | 2022 |
|--------|------|------|------|------|------|------|------|
| NH | 0.364 | 0.352 | 0.426 | 0.393 | 0.390 | 0.420 | 0.660 |
| SQF | 0.369 | 0.350 | 0.401 | 0.442 | 0.471 | 0.564 | 0.543 |
| DBN | 0.285 | 0.301 | 0.352 | 0.465 | 0.530 | 0.533 | 0.511 |
| YLGF | 0.346 | 0.381 | 0.391 | 0.435 | 0.425 | 0.544 | 0.497 |
| HTWY | 0.453 | 0.530 | 0.581 | 0.569 | 0.517 | 0.566 | 0.513 |

**Table 12. 2016–2022 agricultural product supply chain resilience Markov sequence fitting value.**

| Sample | 2016 | 2017 | 2018 | 2019 | 2020 | 2021 | 2022 |
|--------|------|------|------|------|------|------|------|
| NHL | 0.364 | 0.352 | 0.426 | 0.393 | 0.390 | 0.420 | 0.660 |
| SQF | 0.369 | 0.350 | 0.401 | 0.442 | 0.471 | 0.564 | 0.543 |
| DBN | 0.285 | 0.301 | 0.352 | 0.465 | 0.530 | 0.533 | 0.511 |
| YILI | 0.346 | 0.381 | 0.391 | 0.435 | 0.425 | 0.544 | 0.497 |
| HTGF | 0.453 | 0.530 | 0.581 | 0.569 | 0.517 | 0.566 | 0.513 |

**Table 13. 2016–2022 agricultural product supply chain resilience Markov corrected fitting residuals.**

| Sample | 2016 | 2017 | 2018 | 2019 | 2020 | 2021 | 2022 |
|--------|------|------|------|------|------|------|------|
| NHL | 0.016 | 0.006 | 0.016 | 0.006 | 0.014 | 0.018 | 0.061 |
| SQF | 0.008 | 0.009 | 0.001 | 0.001 | 0.006 | 0.010 | 0.000 |
| DBN | 0.001 | 0.006 | 0.010 | 0.009 | 0.006 | 0.001 | 0.015 |
| YILI | 0.001 | 0.018 | 0.009 | 0.001 | 0.010 | 0.016 | 0.009 |
| HTGF | 0.002 | 0.000 | 0.009 | 0.002 | 0.008 | 0.006 | 0.009 |

Note: Residual value is in units of .

The overall model could be acknowledged that the predicted curve could exhibit the negative impact of the pandemic on the agricultural product supply chain resilience developing after 2021. The effect of pandemical disruptions was rigorously stimulated by the model. The general downward during 2024–2025 could be assumed as a delayed impact of the pandemic or the potential occurrence of new public emergencies.

However, Table 11 potentially reflected that the Markov forecast sequences gradually tended to converge their limiting distribution status after multiple transferring steps. Moreover, the experimented modification symbol of different samples in different years was the same. The convergence of the forecasts might be delayed if the amount of data being predicted (explanation variables) is enlarged (Yan,2021). In other words, the real scenario would likely continue the fluctuation trend in the previous years and would be distinct from each other, rather than steadily growing or showing a consistent trend. Therefore, future studies could introduce much more amount of data to derive a more accurate and precise prediction result in the coming years.

**4.3.2. Residual testing of forecasting model fitting effects.** Fu (2022) also proposed a reference level system of a grey forecast model forecasting effect based on previous studies, to further verify the reliability of the improved Markov-modified GM (1,1) model. It is necessary to carry out residual testing, using equation (14), to calculate the grey correlation R, the posterior variance ratio C, and the small probability error P, the results are shown in Table 14.

As can be seen from Table 14, Markov-modified samples could all pass and further were marked as "better" except NHL. The grey correlation R of the improved Markov-modified GM (1,1) for the agricultural product supply chain resilience

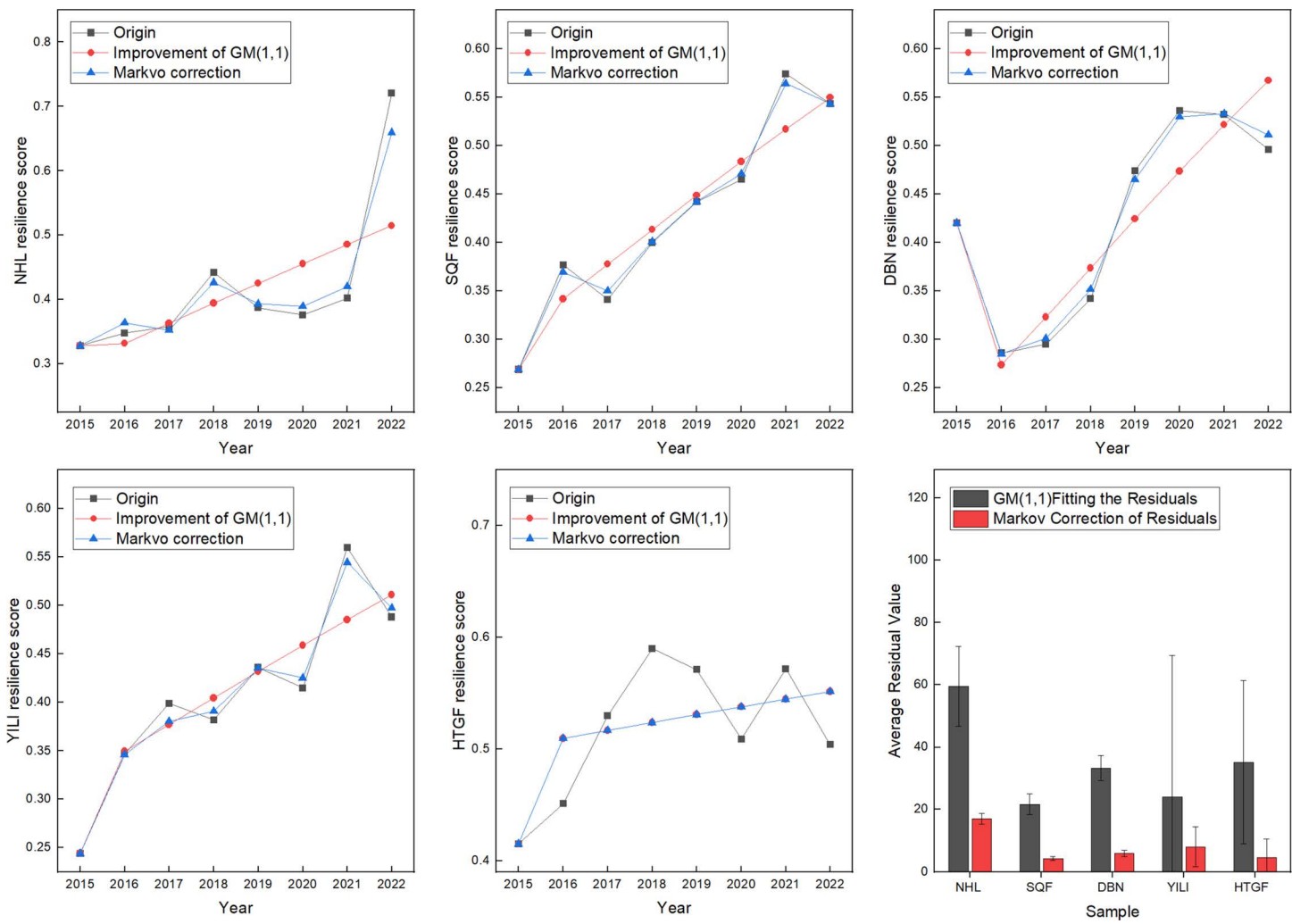

**Fig 3. 2015-2022 the fitting value and average residual value of supply chain resilience for five agricultural product companies.**

**Table 14. Reference level of residual test of improved Markov-modified GM (1,1).**

| Sample | R | C | P | Reference level |
|--------|------|------|------|-----------------|
| NHL | 0.710 | 0.529 | 0.875 | general |
| SQF | 0.641 | 0.183 | 1.000 | better |
| DBN | 0.589 | 0.228 | 1.000 | better |
| YILI | 0.688 | 0.263 | 1.000 | better |
| HTGF | 0.544 | 0.301 | 1.000 | better |

ranges from 0.500 to 0.750. The reference levels demonstrated that Markov could significantly correct the errors deriving from GM (1,1), which verified the great success of the novel improvement from this research. However, the failed sample of Markov modification pointed out that the modification might have some shortcomings. Generally speaking, the result could be considered credible and reliable after the modification.

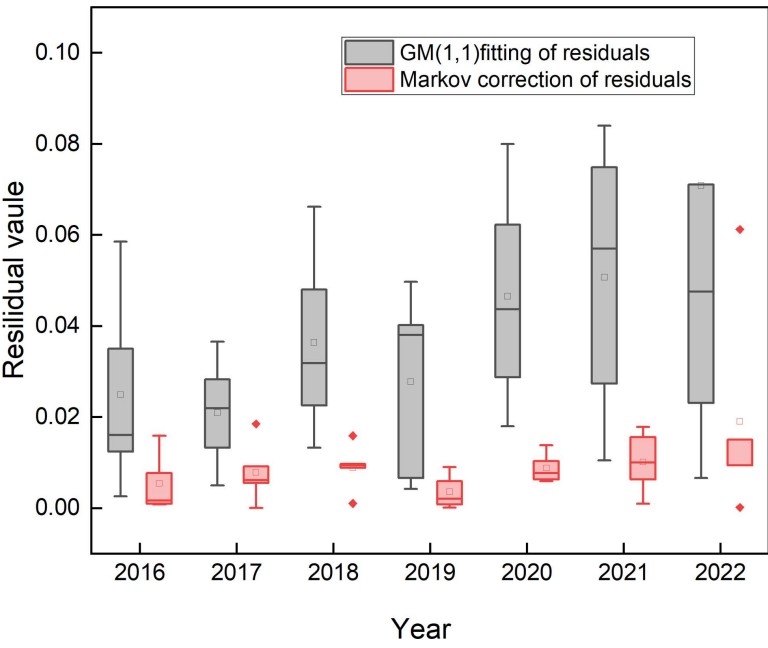

**Fig 4. 2016-2022 prediction residuals of two models.**

Fig 4 shows the distribution state of residuals of the two forecasting models, respectively.

As can be seen from Fig 4, the residual value of the improved GM (1,1)-Markov combination model has lower distributions, and the outliers are closer to the boxes than the GM (1,1) model. It can be seen that the residuals across samples deriving from the GM (1,1) model were distributed between 0.003–0.080, while the ones from the Markov-modified model were between 0.001–0.061, which was much denser than the former. This indicates that the predicted residuals of the GM (1,1) are larger, and the prediction is not stable enough, while the improved GM (1,1)-Markov combination model has smaller overall residuals. Comparing Table 10 and Table 11, it can be seen that after Markov correction, the model overcomes the defects of the grey prediction of the oscillatory series, which makes the fitting effect between the predicted value and the actual reference value more accurate.

**4.3.3. Significant testing of forecasting model statistics traits.** The predicting effect of the resilience forecast models needs to be tested for their statistical significance. Specifically, an independent sample T-test was conducted between the predicted sequences and the original sequences to verify whether there was a significant difference in their means. It could indicate no significant difference in means where the Sig>0.95 in the T-test. The higher the significance, the lower the difference between the means of two sequences. Table 15 shows the T-test result of the two models.

As could be seen from Table 15, the sequences of the predicted value of the two forecasting models under a majority of samples were able to pass the significance test, while the significance of the Markov-modified model was even higher on average. These indicated that the difference between them and the initial resilience evaluation values was very slight. In particular, the test on SQF under the Markov-modified model was very close to 1, indicating that it was almost identical to the raw sequence from 2015 to 2022. This suggested that the model had a surpassing fitting effect. However, there were a small number of samples in the table with reduced significance after the modification, suggesting that the method would have certain deficiencies and less precise when fitting some data distributed in some cases.

**Table 15. Significant test result of two forecasting models.**

| Sample | T Value | | Significance | | Std. Error | |
|---|---|---|---|---|---|---|
| | GM (1,1) | Modified | GM (1,1) | Modified | GM (1,1) | Modified |
| NHL | 0.068 | -0.035 | 0.946 | 0.973 | 0.466 | 0.499 |
| SQF | 0.011 | 0.022 | 0.992 | 0.983 | 0.176 | 0.180 |
| DBN | 0.007 | -0.034 | 0.994 | 0.973 | 0.103 | 0.105 |
| YILI | 0.175 | 0.028 | 0.864 | 0.978 | 0.753 | 0.859 |
| HTGF | 0.056 | 0.061 | 0.956 | 0.952 | 0.382 | 0.394 |

Note: Std. Error Difference is in units of 10-3.

## 4.4. Further prediction

Based on the 2022 supply chain resilience scores as the initial sequence, the improved GM (1,1)-Markov combination prediction method is used to predict the agricultural product supply chain resilience from 2023 to 2027, which solves the Q3 proposed in this dissertation. The results are shown in Table 16 and Fig 5.

As can be seen from Table 14 and Fig 5, the five agricultural product companies' supply chain resilience has shown a fluctuating upward trend since 2023, among which the agricultural product supply chain resilience of NHL agricultural product company decreased from 2023 to 2024. The decline in resilience is due to the company's reorganization, which is a teething period, making the agricultural product supply chain less resilient. In 2026, the agricultural product supply chain resilience of four agricultural product companies (SQF, DBN, YILI, HTGF) occurs a certain degree of decline, while by 2027 begins to rebound. The reasons for this need to be further validated in future research.

According to the prediction results in 2027, the supply chain resilience of the four agricultural product companies (SQF, DBN, YILI, HTGF) is higher than 0.600, reaching a very high level. Among these, the agricultural product supply chain resilience of DBN in 2027 is more than 0.750, which is the highest among the five companies. From Table 15, it can also be concluded that over the next five years, the supply chain resilience of NHL agricultural product company is expected to increase by 25%, the supply chain resilience of SQF agricultural product company is expected to increase by 26%, the supply chain resilience of DBN agricultural product company is expected to increase by 45%, the supply chain resilience of YILI agricultural product company is expected to increase by 24%, and the supply chain resilience of HTGF agricultural product company is expected to increase by 10%. It can be assumed that after experiencing major public emergencies, although listed agricultural product companies are affected by the public emergencies in the short term, the impact of crisis events on supply chain resilience will diminish, recover gradually from the crisis, and benefit from the crisis for the better in the long run.

## 4.5. Sensitivity analysis

This research work, using the Solomon case, evaluates the impact of inventory turnover rate (ITR), supply chain operating cost (SCOC), and supplier spatial distance (SSD) on supply chain resilience through sensitivity analysis.

**Table 16. 2023–2027 forecasted value of agricultural product supply chain resilience.**

| Sample | 2023 | 2024 | 2025 | 2026 | 2027 |
|---|---|---|---|---|---|
| NHL | 0.609 | 0.505 | 0.532 | 0.689 | 0.713 |
| SQF | 0.569 | 0.618 | 0.652 | 0.654 | 0.686 |
| DBN | 0.609 | 0.672 | 0.709 | 0.665 | 0.741 |
| YILI | 0.523 | 0.575 | 0.599 | 0.594 | 0.616 |
| HTGF | 0.537 | 0.586 | 0.593 | 0.557 | 0.564 |

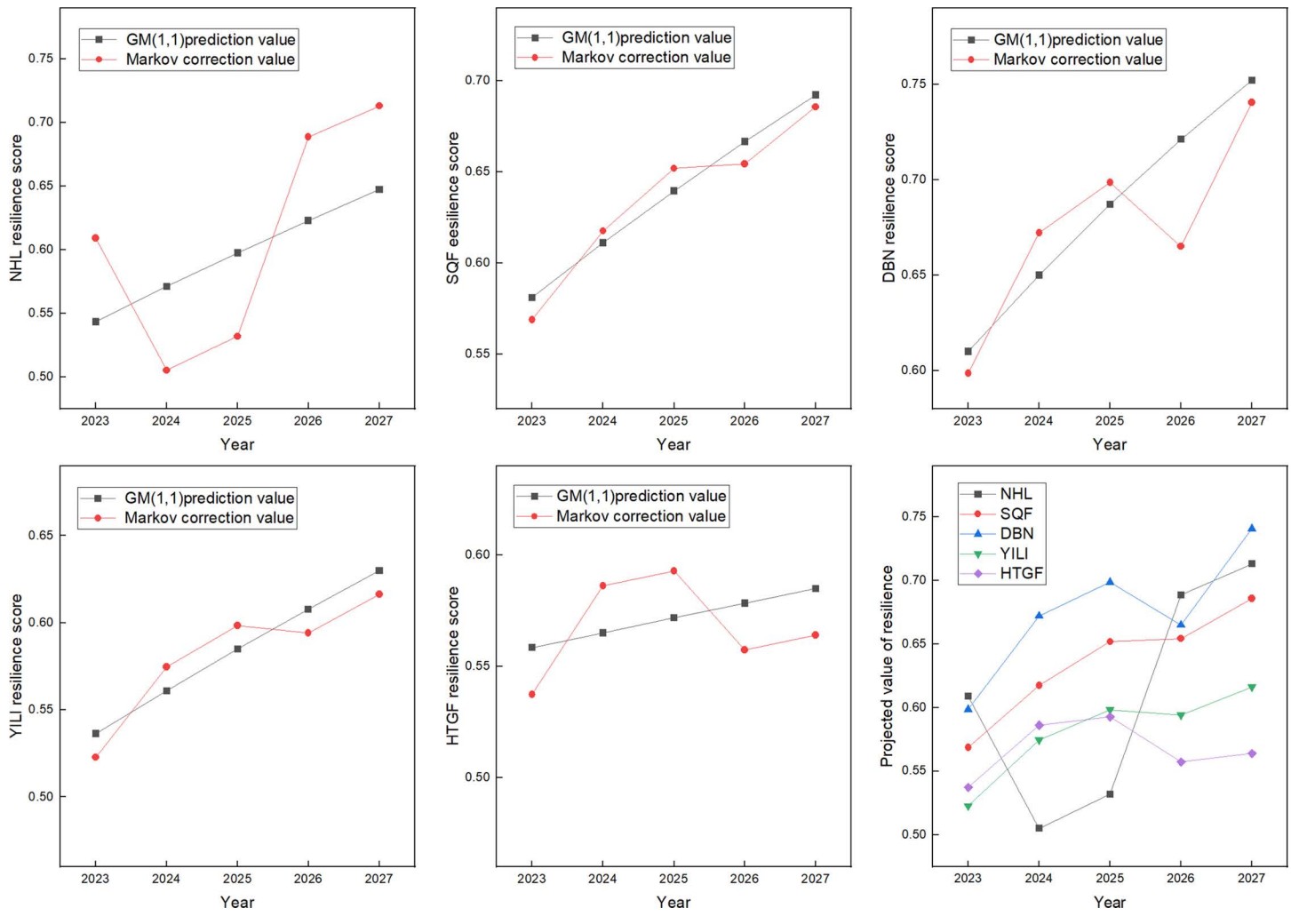

**Fig 5. 2023-2027 trend of supply chain resilience for five agricultural product companies.**

**4.5.1. Sensitivity analysis of inventory turnover rate.** This dissertation takes the top five agricultural product listed companies in China as examples, setting the inventory turnover frequency to 10, 15, 20, and 25 times respectively, running formula (5) to conduct sensitivity experiment tests, and analyzing the impact of inventory turnover rate on the resilience of the agricultural product supply chain. The results are compared as shown in Fig 6. The comprehensive case results reveal that as the supplier's inventory turnover rate continues to increase, the resilience of the supply chain also continues to improve; when the supplier's inventory turnover rate changes from 5 times to 10 times, the improvement in supply chain resilience is relatively small; when the supplier's inventory turnover rate changes from 15 times to 20 times, and from 20 times to 25 times, the improvement in supply chain resilience is more significant. The analysis suggests that a high inventory turnover rate means that suppliers can quickly respond to market changes and order demands. Especially under public emergencies, this rapid response capability is an important indicator of supply chain resilience, as it allows suppliers to quickly adjust in the face of market fluctuations, thereby reducing the risk of supply chain disruptions. By analyzing the data from different cases, it is found that due to the relatively dispersed distribution of NHL's suppliers, while the other four agricultural product listed companies have a more concentrated distribution of suppliers, when the inventory turnover rate

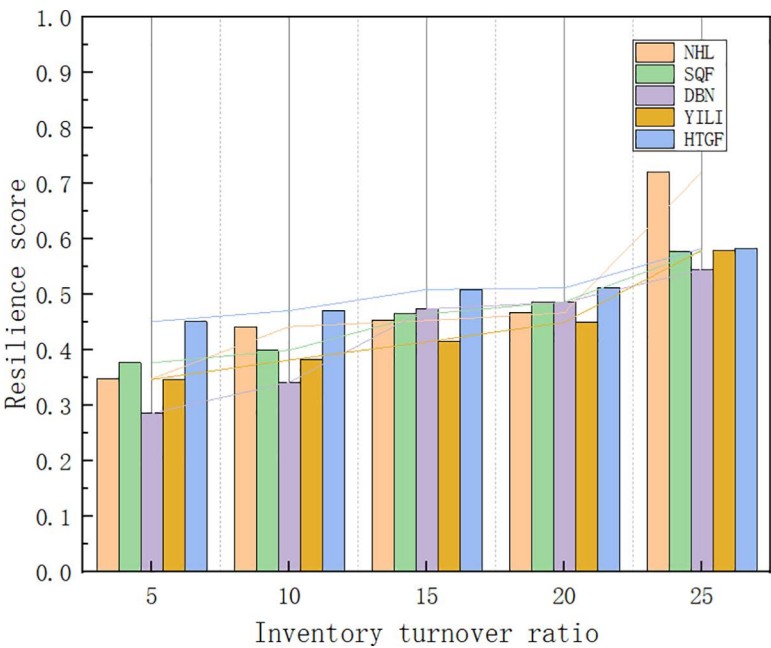

**Fig 6. Sensitivity analysis of inventory turnover rate.**

of agricultural products continuously improves, the increase in the resilience of NHL's supply chain is more significant. In contrast, the increase in the resilience of the supply chains of SQF, DBN, YILI, and HTGF is relatively smaller.

**4.5.2. Sensitivity analysis of supply chain operating cost.** In this study, the operating costs of the agricultural product supply chain were set at 100,000 yuan/year, 150,000 yuan/year, 200,000 yuan/year, and 250,000 yuan/year, respectively. Using formula (6), sensitivity experiments were conducted to analyze the impact of agricultural product suppliers' operating costs on the resilience of the supply chain. The comparative results are shown in Fig 7.

From the comprehensive case study results, it can be observed that as supply chain operating costs continue to rise, supply chain resilience decreases. When supply chain operating costs increased from 50,000 yuan/year to 100,000 yuan/year, the decline in supply chain resilience was significant; whereas, when costs rose from 150,000 yuan/year to 200,000 yuan/year, and from 200,000 yuan/year to 250,000 yuan/year, the decrease in resilience was less pronounced. The reasons for this include: On one hand, during the COVID-19 pandemic, lockdown measures and border closures in various countries and regions led to logistical disruptions and a significant increase in transportation costs. Delays and increased expenses in air and sea transportation, as well as traffic controls for land transport, made the transportation of agricultural products more difficult and expensive. To address these challenges, listed agricultural companies had to bear higher transportation costs, which not only increased operating costs but also reduced the flexibility and responsiveness of the supply chain. On the other hand, the uncertainty of market demand in the post-pandemic period has increased. To cope with sudden changes in supply and demand, companies had to increase inventory levels, leading to higher storage and management costs. However, excessive inventory not only ties up a large amount of capital but can also lead to waste due to the shelf-life limitations of agricultural products. The increased cost of inventory management further weakened the financial condition of enterprises. Therefore, as the investment in supply chain operating costs increases, supply chain resilience is further reduced.

**4.5.3. Sensitivity analysis of supplier spatial distance.** This dissertation selects five agricultural product listed companies and sets the spatial distances of agricultural product suppliers at 10km, 15km, 20km, and 25km respectively.

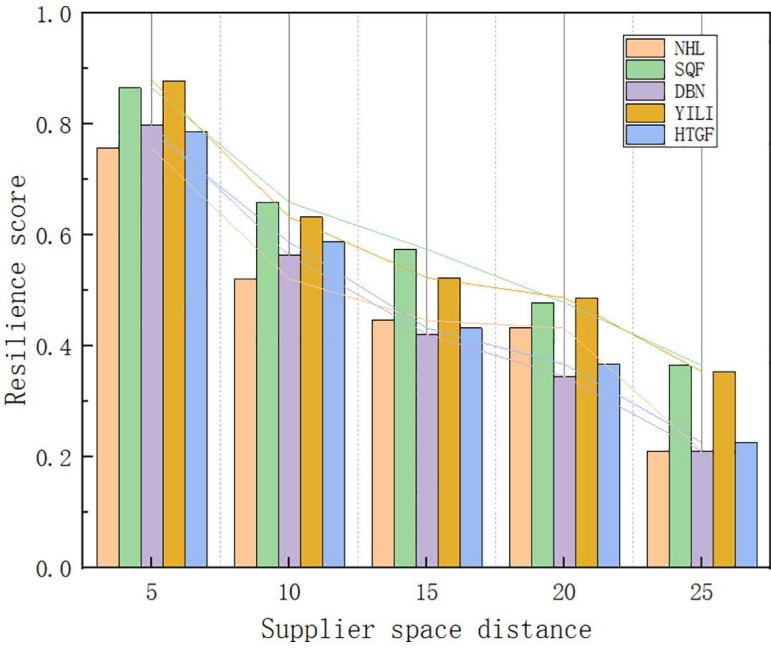

**Fig 7. Sensitivity analysis of supply chain operational cost.**

By running formula (7), sensitivity experiments are conducted to analyze the impact of supplier spatial distance on the resilience of the supply chain. The comparison of the results is shown in Fig 8.

As shown in Fig 8, with the increasing spatial distance of agricultural product suppliers, the resilience of the supply chain continues to decline. When the supplier's spatial distance changes from 10 kilometers to 15 kilometers, the decrease in supply chain resilience is relatively small. However, when the supplier's spatial distance changes from 15 kilometers to 20 kilometers, and then from 20 kilometers to 25 kilometers, the decline in supply chain resilience is more significant. The analysis suggests the following reasons: on one hand, the increase in the supplier's spatial distance lengthens the transportation time, resulting in a longer transportation cycle for agricultural products from the production site to the consumption site. This not only increases logistics costs but also reduces the response speed of the supply chain. On the other hand, the increase in distance leads to more supply chain nodes (such as transfer warehouses, logistics hubs, etc.), and each node can potentially become a bottleneck or risk point. Therefore, a larger supplier spatial distance is detrimental to the improvement of supply chain resilience.

## 4.6. Discussion

Relevant research scholars (Yan et al., 2021; Qi et al., 2022) have conducted studies on the probability of supply chain risks and the level of vulnerability, mainly focusing on rough measurement and prediction of supply chain resilience. Distinguished from existing studies, this dissertation constructs a supply chain resilience evaluation model tailored to the characteristics of the agricultural products industry by adopting an improved EW-TOPSIS. Based on the scientific construction of the evaluation index system, it effectively overcomes the unreasonable impact on index weights caused by over-concentrated or overly sparse sample data distribution under some indicators, obtaining reasonable weights for each index, and deriving the supply chain resilience scores of five leading listed companies in the agricultural products industry. Meanwhile, an improved Markov-modified GM (1,1) model is employed to address the poor prediction effect of traditional grey models for oscillating sequences, achieving high-precision fitting of the existing development trends of agricultural

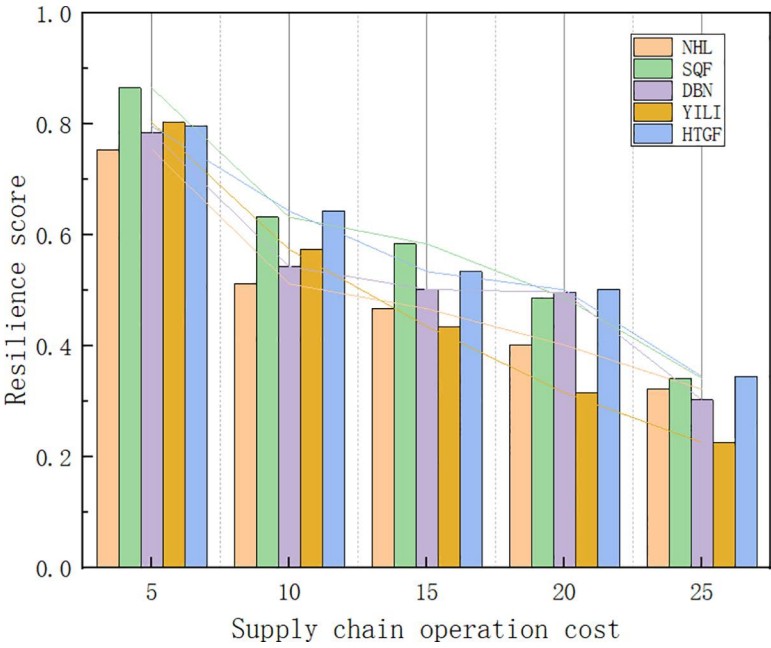

**Fig 8. Sensitivity analysis of supplier spatial distance.**

product supply chain resilience, and deriving relatively credible trend prediction values for the next five years. Therefore, compared with existing research results, this study can more accurately and reasonably measure and predict the resilience of agricultural product supply chains, providing reliable references for the overall development trends of China's agricultural product supply chains.

Furthermore, the specific analysis is shown below:

First of all, an enterprise with a higher market capitalization, a large size, or strong profitability would not exhibit strong supply chain resilience. The results of the study showed that the ranking of market value is not consistent with the ranking of resilience level. For example, the NHL owed the highest market value in 2022 among all the industries but showed a significant lack of resilience during the pandemic(Q2). Secondly, in the evaluation system, indicators such as delayed delivery rate, suppliers' cash flow ratio, and inventory turnover ratio have higher weightings, indicating their greater contribution to supply chain resilience; under the backdrop of public crisis events, enterprises need to seek communication and collaboration more, through more open and transparent information sharing and cooperation mechanisms to jointly face difficulties, and work together with member enterprises in the supply chain to enhance resilience and corporate strength(Q1). All the enterprises along the domestic agricultural supply chain have to cohesion through cooperation altogether to prevent and defuse potential public health emergencies in the future. Thirdly, in the prediction of empirical results, it was found that all five companies had their best performance in 2020; primarily because in 2020, the government's support for ensuring supplies was strong, and a few companies with ample inventory (mainly leading listed companies) quickly utilized emergency logistics to sell their products; however, performance alone cannot fully indicate supply chain resilience, and in fact, the resilience scores of these companies in 2020 were relatively low, mainly due to the significant impact of negative indicators such as logistics efficiency and delayed deliveries, and the ability to quickly liquidate inventory does not entirely represent supply chain resilience. By predicting the development trend of agricultural product supply chain resilience for sample enterprises from 2023 to 2027, it shows that under the background of the weakening impact of the pandemic, the resilience of each enterprise is gradually and steadily improving (Q3). Finally, this dissertation

combines the Solomon case study to evaluate the impact of inventory turnover rate (ITR), supply chain operating cost (SCOC), and supplier spatial distance (SSD) on supply chain resilience through sensitivity analysis. The case study analysis reveals that, under public emergencies, agricultural product listed companies should increase the inventory turnover rate, reduce supply chain operating costs, and shorten the spatial distance between suppliers, in order to better cope with demand fluctuations within each link of the supply chain.

The methods and results implemented in this article will provide guiding significance for the selected case study in the following aspects. Firstly, this research adopted an improved EW-TOPSIS method to construct an agricultural supply chain resilience evaluation system in line with the traits of the agri-food industry. The method overcame the unreasonable disturbance of overly dense or sparse distribution of sampled data, and derived the reasonable proportion of indicators, influence factors of the supply chain of which is selected by a scientific principle, and assessed the resilience level of five top listed agricultural companies. An improved Markov-modified GM (1,1) forecast model was established in this research, which overcame the deficiency of the predictive effect on oscillatory sequences and achieved a highly fitted effect of the development tendency of five sampled companies' supply chain resilience level. Secondly, based on the research findings of evaluation and forecasting models in this dissertation, improvement mechanisms are proposed from multiple levels such as government, industry, and enterprises to effectively address the issues in the selected case studies (Q4):

(1) **Government subsidy mechanisms.** Government subsidies are an important factor in ensuring the stable operation of the agricultural product supply chain during emergencies. Among the various methods the government uses to regulate the macroeconomic context and ensure the smooth operation of major livelihood industries, economic subsidies are the most direct and effective. The government should proactively conduct grassroots research to understand the situation of enterprises in difficulty and formulate subsidy programs that comprehensively grasp the damage to enterprises immediately after the occurrence of emergencies. On one hand, accelerate the delivery of payments for government procurement in which enterprises participate, and appropriately extend or reduce the payment of taxes and social security funds by enterprises; on the other hand, encourage financial institutions such as state-owned banks to relax financing and loan limits for enterprises in distress, thus injecting funds into the market to boost market confidence and provide direct financial support for enterprise production.

(2) **Development of digital supply chain mechanisms**. In recent years, the digital transformation of large enterprises has been a significant topic in the field of enterprise management. Table 5 shows that the delayed delivery rate, supplier cash flow ratio, and inventory turnover rate account for 14.00%, 10.60%, and 8.00%, respectively. The construction and development of digital supply chains can directly influence the efficiency and effectiveness of inter-enterprise collaboration and enhance the transparency of supply chain information. As leading enterprises in the industry, five companies should take the initiative to play a leading and driving role in building digital supply chains, for example, YILI has a large number of suppliers and should actively construct an intelligent supplier model, focusing on timely control during the change period of suppliers, reducing the time for interest coordination between various departments in agriculture, and enhancing agriculture's rapid response ability and regulatory capacity toward suppliers in major emergencies; DBN, with a large number of suppliers and relatively stable supply relationships, should effectively coordinate among suppliers in various forms, forming multi-channel partnerships and promoting information resource sharing and complementary advantages. SQF, maintaining its leading position in the agricultural sector as a listed company, needs to build a digital management system to coordinate internal management with external supply chain management to maintain market share and prevent potential crises of "large enterprise diseases"; HTGF and NHL should actively build digital supply chain management platforms, optimize internal and external management efficiency, and enhance decision-making and management capabilities using large-scale data.

(3) **Innovation in multi-source procurement channel mechanisms.** As shown in Table 6, the dimension of agricultural product supply chain regulation capability accounts for 31.10% of the evaluation index system, making it one

of the most important dimensions determining the resilience of the agricultural product supply chain. Therefore, the agricultural product supply chain industry should comprehensively build diversified supply channels for agricultural enterprises, connecting offline and online networks to form a unified supply approach, strengthening the cooperative relationships with agricultural processing enterprises, and standardizing production environments, processing, and acceptance processes. NHL, primarily engaged in seed business, has a complete system for seed research, development, and sales, which provides a good condition for constructing self-cultivated and self-sold agricultural bases. It can expand internal procurement, turnover, and supply and delivery models, implementing intelligent control and integrating them comprehensively into enterprise management. SQF and HTGF, with a wide variety of product types and dispersed procurement sources, should establish their own procurement platforms, screen high-quality suppliers through online bidding channels, and periodically evaluate the performance of upstream enterprises in the supply chain, selecting high-quality enterprises for long-term cooperation, gradually perfecting a fixed multi-source procurement system; YILI and DBN can enhance cooperation and links with upstream enterprises and improve the resilience of the supply chain embodied in the procurement process by building online and offline supply networks, unifying procurement channels, and standardizing procurement procedures and logistics systems.

(4) **Improvement of emergency response mechanisms for sudden events.** Defining the work areas, service items, operating conditions, working methods, cooperation principles, and supervision aspects of various units helps to mitigate the risks brought about by sudden events. Although YILI and HTGF are large-scale enterprises, they exhibited insufficient resilience during the pandemic and should comprehensively consider the local characteristics, natural and geographical environments, production and personnel allocation, social productivity distribution, and actual transportation conditions of their subsidiaries in various locations. Under the guidance and support of governments at all levels, they should establish a three-level supply network with "district-town (street, committee)-community (village)" that is longitudinally connected, horizontally supported, and appropriately scaled; DBN, whose main suppliers are agricultural production cooperatives, is suitable for promoting digital reform in emergency materials in rural areas, establishing a "village-town (street)-district" three-level linkage mechanism. SQF and NHL should focus on emergency material digital platforms, improving coverage and inventory rates, and ensuring scientific, effective, and timely management of the entire process of material reserves, strengthening the allocation of materials for emergency response.

The management and practical significance of this work are as follows. On the one hand, this research will assist governments and stakeholders in other countries in understanding the key elements of supply chains and strengthening supply chain resilience. Identifying core elements can help managers find the strengths and weaknesses of supply chains. By focusing on relevant indicators, they will determine the effectiveness of existing supply chain management strategies. Additionally, they can learn from major public emergencies to establish mechanisms for pre-event prevention, in-event adjustment, and post-event feedback, thereby promoting the optimization and sustainable development of global agricultural product supply chains. On the other hand, this research identifies "agricultural product supply chain adjustment capabilities (A3)" as the key determinant to achieve resilience in agricultural product supply chains, warranting the highest priority. The COVID-19 pandemic has highlighted the importance of adjustment capabilities for agricultural product-listed companies in building resilient supply chains. The agricultural product supply chain faces numerous challenges, such as natural disasters, market fluctuations, policy changes, etc., which can all impact the supply chain. Supply chains with strong adjustment capabilities can manage inventory more effectively, reducing overstocking or stockouts. Therefore, in this context, adjustment capability is not only a theoretical concept but also a practical necessity for companies to survive and thrive in times of crisis, directly relating to the company's profitability and market competitiveness. All stakeholders should pay attention to the effective coordination of various links in the agricultural product supply chain, such as delayed delivery rate (B12), supplier cash flow ratio (B7), inventory turnover rate (B6), etc. On one hand, enterprises upstream and downstream of the supply chain should build information-sharing mechanisms, such as data on market supply and

demand, price fluctuations, and logistics status, to improve the overall efficiency of the supply chain. On the other hand, long-term strategic cooperative relationships should be established with suppliers and customers to ensure the stability and sustainability of the supply chain. A reasonable interest distribution mechanism should be formulated to incentivize participants at all links of the supply chain to jointly maintain its resilience.

## 5. Conclusions

This article is based on the research background of public emergencies. On the one hand, five typical agricultural product listed companies from 2015 to 2022 were selected as research samples. This article constructed an agricultural product supply chain resilience index system covering four dimensions: receptivity, adaptability, resistance, and recovery, and used an improved EW-TOPSIS model to scientifically measure the resilience of their agricultural product supply chains. On the other hand, the improved GM (1,1)-Markov model proposed in this dissertation was used to accurately predict the development trend of the agricultural product supply chain resilience of the five agricultural product companies from 2023 to 2027.

### 5.1. Findings

The main findings of this study are as follows: Firstly, this research work divides the evaluation indicators of agricultural product supply chain resilience from both quantitative and qualitative perspectives, establishing a more complete evaluation system for the supply chain resilience of agricultural enterprises. In the improved entropy weight-TOPSIS model, the introduction of Z-score parameters effectively overcame the unreasonable impact on the weight of indicators caused by the overly dense or sparse distribution of sample data under some indicators, obtained reasonable weights for each indicator, and derived the supply chain resilience scores of the top five agricultural product listed companies, improving the accuracy of the evaluation results. Through empirical analysis, the primary influencing factors of agricultural product supply chain resilience were identified, explaining the importance of indicators such as the delayed delivery rate, supplier cash flow ratio, and inventory turnover rate for enterprise supply chain resilience(Q1); it also demonstrated a general trend where the level of agricultural product supply chain resilience declined in 2020 and 2021 and recovered and reached its highest point in 2022(Q2).

Secondly, using the top agricultural product listed companies before and after the outbreak of the COVID-19 pandemic over eight years as samples, a resilience prediction model was constructed using the improved GM (1,1)-Markov prediction method to predict the development trends in the resilience of the agricultural product supply chains of the sample companies from 2023 to 2027. It presented a trend of gradual and stable improvement in resilience as the impact of the COVID-19 pandemic gradually diminishes(Q3). The resilience prediction model proposed in this study can predict changes in resilience indicators in real-time, providing assistance in the evaluation and early warning of supply chain resilience of agricultural product listed companies, thereby reducing the risk of chain breakage and blockage in uncertain environments. However, the limitation of this study lies in its dependence on the collection of financial indicator data from agricultural product listed companies. Therefore, future research should focus on acquiring the required data through multiple channels.

Lastly, the prediction effectiveness of the resilience prediction model needs to be tested for significance. By conducting independent samples T-tests on the predicted series and the original series, the mean difference was compared to determine if it was significant. The results found that at the significant level, there was no significant difference between the GM (1,1) prediction series and the Markov corrected series; the prediction effects of the five agricultural product listed companies passed the significance test with high significance and small mean differences, thereby verifying the feasibility of the improved GM (1,1)-Markov. In addition, this dissertation combines the Solomon case study to evaluate the impact of inventory turnover rate (ITR), supply chain operating costs (SCOC), and supplier spatial distance (SSD) on supply chain resilience through sensitivity analysis. The results show that improving inventory turnover rate, reducing

operating costs, and shortening supplier spatial distance are key measures to enhance supply chain resilience. Meanwhile, in combination with the case background of the five agricultural product listed companies and the results obtained from empirical research, five categories of agricultural product supply chain enhancement mechanisms are proposed at the government level, industry level, and enterprise level, aiming to provide reference for the future development of relevant fields(Q4).

### 5.2. Limitations and prospect

Like other studies, this study has some limitations: (1) Errors could occur due to these data sources being major relevant to the financial performances of the sampled company, which enlarged the effect of finance on the resilience system. The data of objective quantitative indicators are collected from publicly disclosed annual financial reports of sample companies, which leads to a significant impact on the economic status of sample companies on supply chain resilience. (2) from the perspective of the model methods. The improved GM (1,1)-Markov model is significantly influenced by the scale of the data. while this research only gathered 15 indicators of 5 samples during 8 years, 600 records of data. This could result in a small scale of the state space in the Markov process and the forecast model could not completely stimulate the complexity of external or internal changes in the real environment.

In future research. On one hand, future studies are expected to formulate a more complex, integrated, and sufficient indicator system to assess the resilience level more systematically. Deeper research would apply the metaheuristic approaches to derive more explicit, detailed, and actual estimates of the agricultural product supply chain resilience. On the other hand, in future predictive studies with longer time spans, an increase in the number of sample time points will help enhance the model's accuracy in predicting future time periods. Furthermore, based on the empirical results of supply chain resilience in this study, targeted research on supply chains in other industries can be conducted, and relevant recommendations can be proposed.

### Acknowledgments

The authors would like to thank the anonymous reviewers and editors for their insightful and constructive comments on our paper.

### Author contributions

**Conceptualization:** Hongzhi Wang, Li Lu, Zhaoli Liu, Yuxuan Sun.

**Data curation:** Hongzhi Wang, Li Lu.

**Formal analysis:** Hongzhi Wang, Li Lu, Zhaoli Liu, Yuxuan Sun.

**Funding acquisition:** Hongzhi Wang.

**Investigation:** Hongzhi Wang, Li Lu.

**Methodology:** Hongzhi Wang, Li Lu.

**Resources:** Hongzhi Wang.

**Software:** Li Lu, Zhaoli Liu, Yuxuan Sun.

**Supervision:** Li Lu, Zhaoli Liu, Yuxuan Sun.

**Validation:** Zhaoli Liu, Yuxuan Sun.

**Visualization:** Zhaoli Liu, Yuxuan Sun.

**Writing – original draft:** Hongzhi Wang, Li Lu, Zhaoli Liu, Yuxuan Sun.

**Writing – review & editing:** Hongzhi Wang, Li Lu, Zhaoli Liu, Yuxuan Sun.

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
