## [Decision Letter · Decision Letter 0]

15 Dec 2024

PONE-D-24-26636Study on measurement and prediction of agricultural products supply chain resilience based on improved EW-TOPSIS and GM (1,1)-Markov models under public emergenciesPLOS ONE

Dear Dr. Lu,

Thank you for submitting your manuscript to PLOS ONE. After careful consideration, we feel that it has merit but does not fully meet PLOS ONE’s publication criteria as it currently stands. Therefore, we invite you to submit a revised version of the manuscript that addresses the points raised during the review process.

We look forward to receiving your revised manuscript.

Kind regards,

Nhat-Luong Nhieu, Ph.D.

Academic Editor

PLOS ONE

Journal Requirements:

Reviewers' comments:

Reviewer's Responses to Questions

**Comments to the Author**

1. Is the manuscript technically sound, and do the data support the conclusions?

Reviewer #1: Partly

Reviewer #2: Yes

2. Has the statistical analysis been performed appropriately and rigorously? 

Reviewer #1: No

Reviewer #2: Yes

3. Have the authors made all data underlying the findings in their manuscript fully available?

Reviewer #1: No

Reviewer #2: No

4. Is the manuscript presented in an intelligible fashion and written in standard English?

Reviewer #1: No

Reviewer #2: Yes

5. Review Comments to the Author

Reviewer #1: Your research work is comprehensive and addresses a critical issue. Here are some suggestions to improve it further:

1. Specify the study's geographical and temporal scope early in the introduction.

2. Emphasize why food security and supply chain resilience are crucial, not just for China but globally.

3. Include more references to previous food security and supply chain resilience studies to provide a solid theoretical foundation.

4. Clearly state the gaps in the existing literature the research aims to fill.

5. Provide a brief explanation of the EW-TOPSIS and GM (1,1)-Markov models for readers unfamiliar with these methods.

6. Explain why these models were chosen over others and how they are particularly suited to your research.

7. Use graphs and charts to illustrate key findings, such as the resilience development curve and the predicted trends.

8. Acknowledge any limitations in your data or methodology and how they might affect the results.

9. Compare the findings with those of other studies to highlight similarities or differences.

10. Discuss the broader policy implications of your findings in more detail, suggesting specific actions that policymakers could take.

11. Provide a concise summary of your main findings and their significance.

12. Suggest areas for future research to build on the proposed work.

13. Simplify complex sentences and avoid jargon to make your work more accessible.

14. Ensure there are no grammatical errors or typos.

15. Add software details and computing codes to verify this research results quickly.

16. Research questions/hypotheses of the work are to be discussed in this work.

By incorporating these suggestions, you can enhance the clarity, depth, and impact of your research.

Reviewer #2: This work focused on measurement and prediction of agricultural products supply chain resilience using improved EW-TOPSIS and GM (1,1)-Markov. The topic is interesting, but there are essential concerns about revising this work considerably. The following comments are suggested:

1. The research gap and contribution are not clear in the Abstract. What problem did you study, and why is it important? The authors just mentioned the general issues in the supply chain. What is the main problem of the case study?

2. An updated and complete literature review should be conducted to present the state-of-the-art and knowledge gaps in the research with strong relevance to the paper's topic. Some recent leading works have focused on optimizing the supply chain using optimization approaches such as metaheuristic approaches.

3. I recommend extending the background on optimization methods, providing a thorough justification for the choice of method, and conducting a comprehensive sensitivity analysis using recommended approaches.

4. A robust and comprehensive section is necessary for their Data Collection. How do authors collect their data? What are the sources of data?

5. A detailed explanation is required for the case study—the location of suppliers, the distances, type of transportation, unit cost for transport, holding, storage, and so on.

6. Validation and verification are the main parts of the modeling approach. Please add a separate section to show the validation of the developed model.

7. A separate strong discussion before the conclusion should be revised to discuss the results and compare them with similar studies. What are the managerial and practical implications of this work? How do the implemented approach and results help the selected case study? All details should be discussed and organized well.

8. What is this work's managerial and practical implication for the selected case study? How could these results help the governments and stakeholders in other countries?

9. What is the generalizability aspect of this work compared to other similar SC worldwide?

10. The conclusion is very general. The conclusion should be revised and improved by adding this work's significant contribution, results, and limitations.

11. The structure of paper (just 5 sections are needed) should be revised. Introduction-Methodology (data collection-case study)-Literature review- results and discussion- Conclusion (all sub-sections (6.1-6.3) should be transferred to the discussion part).

6. PLOS authors have the option to publish the peer review history of their article (what does this mean? ). If published, this will include your full peer review and any attached files.

**Do you want your identity to be public for this peer review?** For information about this choice, including consent withdrawal, please see our Privacy Policy .

Reviewer #1: **Yes: ** Dr. Irfan Ali, M.Phil., Ph.D.

Reviewer #2: No

---

## [Author Response · Author response to Decision Letter 1]

15 Jan 2025

Thank you for your letter and for the reviewers’ comments concerning our manuscript entitled “Study on measurement and prediction of agricultural product supply chain resilience based on improved EW-TOPSIS and GM (1,1)-Markov models under public emergencies (ID: PONE-D-24-26636)”. These comments are valuable for the revision and improvement of our paper and hold significant guiding significance for our research. We have carefully studied the feedback and made corrections. We hope for your approval. The revised sections are marked in yellow on the paper. The main corrections made to the paper and our responses to the reviewers’ comments are detailed below.

---

## [Decision Letter · Decision Letter 1]

4 Mar 2025

Study on measurement and prediction of agricultural product supply chain resilience based on improved EW-TOPSIS and GM (1,1)-Markov models under public emergencies

PONE-D-24-26636R1

Dear Dr. Lu,

We’re pleased to inform you that your manuscript has been judged scientifically suitable for publication and will be formally accepted for publication once it meets all outstanding technical requirements.

Kind regards,

Nhat-Luong Nhieu, Ph.D.

Academic Editor

PLOS ONE

Additional Editor Comments (optional):

Reviewers' comments:

Reviewer's Responses to Questions

**Comments to the Author**

1. If the authors have adequately addressed your comments raised in a previous round of review and you feel that this manuscript is now acceptable for publication, you may indicate that here to bypass the “Comments to the Author” section, enter your conflict of interest statement in the “Confidential to Editor” section, and submit your "Accept" recommendation.

Reviewer #1: All comments have been addressed

Reviewer #2: (No Response)

Reviewer #3: All comments have been addressed

2. Is the manuscript technically sound, and do the data support the conclusions?

Reviewer #1: Yes

Reviewer #2: (No Response)

Reviewer #3: Yes

3. Has the statistical analysis been performed appropriately and rigorously? 

Reviewer #1: Yes

Reviewer #2: (No Response)

Reviewer #3: Yes

4. Have the authors made all data underlying the findings in their manuscript fully available?

Reviewer #1: Yes

Reviewer #2: (No Response)

Reviewer #3: Yes

5. Is the manuscript presented in an intelligible fashion and written in standard English?

Reviewer #1: Yes

Reviewer #2: (No Response)

Reviewer #3: Yes

6. Review Comments to the Author

Reviewer #1: I found the authors addressed all the comments and incorporated all the suggestions in the revision.

Reviewer #2: After reviewing the revised version of the manuscript, I have noted that the authors have properly addressed the indicated comments in my previous revision. Therefore, I recommend to accept this new version of the manuscript.

Reviewer #3: In the introduction, highlight more explicitly why resilience measurement and prediction are crucial under public emergencies (e.g., COVID-19, African Swine Fever).

The authors give a good summary of prior methods (e.g., fuzzy AHP, SCOR, etc.). Compare the proposed approach with these methods more systematically. This helps justify why the combination of EW-TOPSIS and GM(1,1)-Markov is particularly apt for resilience measurement and forecasting.

The paper lists a set of resilience indicators. While table 4 is clear, it would help to briefly explain/describe each indicator.

Briefly explore possible causes—e.g., different pandemic-control policies in certain regions, the firm’s digital advantage, or special government support - for the sudden increased in resilience evaluation score of NHL from 2021-2022.

As your work deals with food security and major emergencies, there may be social or policy implications. Authors can consider to provide more recommendations based on these implications.

The manuscript is mostly clear, but do a quick proofread to ensure no grammar mistake is left.

7. PLOS authors have the option to publish the peer review history of their article (what does this mean? ). If published, this will include your full peer review and any attached files.

**Do you want your identity to be public for this peer review?** For information about this choice, including consent withdrawal, please see our Privacy Policy .

Reviewer #1: **Yes: ** Irfan Ali

Reviewer #2: No

Reviewer #3: No

---

## [Editor Report · Acceptance letter]

PONE-D-24-26636R1

PLOS ONE

Dear Dr. Lu,

I'm pleased to inform you that your manuscript has been deemed suitable for publication in PLOS ONE. Congratulations! Your manuscript is now being handed over to our production team.

Kind regards,

on behalf of

Asst. Prof. Nhat-Luong Nhieu

Academic Editor

PLOS ONE